# Dynamics of replication origin over-activation

Haiqing Fu[1], Christophe E. Redon[1], Bhushan L. Thakur[1], Koichi Utani[1,2], Robin Sebastian  [1], Sang-Min Jang[1,3], Jacob M. Gross[1], Sara Mosavarpour[1], Anna B. Marks[1], Sophie Z. Zhuang[1], Sarah B. Lazar[1], Mishal Rao[1], Shira T. Mencer[1], Adrian M. Baris[1,4], Lorinc S. Pongor [1] & Mirit I. Aladjem [1✉]

Safeguards against excess DNA replication are often dysregulated in cancer, and driving cancer cells towards over-replication is a promising therapeutic strategy. We determined DNA synthesis patterns in cancer cells undergoing partial genome re-replication due to perturbed regulatory interactions (re-replicating cells). These cells exhibited slow replication, increased frequency of replication initiation events, and a skewed initiation pattern that preferentially reactivated early-replicating origins. Unlike in cells exposed to replication stress, which activated a novel group of hitherto unutilized (dormant) replication origins, the preferred re-replicating origins arose from the same pool of potential origins as those activated during normal growth. Mechanistically, the skewed initiation pattern reflected a disproportionate distribution of pre-replication complexes on distinct regions of licensed chromatin prior to replication. This distinct pattern suggests that circumventing the strong inhibitory interactions that normally prevent excess DNA synthesis can occur via at least two pathways, each activating a distinct set of replication origins.

[1] Developmental Therapeutics Branch, Center for Cancer Research, National Cancer Institute, Bethesda, MD, USA. [2] Present address: Department of Microbiology, Kanazawa Medical University, Uchinada Ishikawa, Kanazawa, Japan. [3] Present address: Department of Biochemistry, Chungbuk National University, Chungdae-ro, Seowon-gu, Cheongju, Korea. [4] Present address: Program in Cancer Biology, Oregon Health and Science University, Portland, OR, USA. ✉email: aladjemm@mail.nih.gov

Proliferating cells ensure the accurate transmission of their genetic material to daughter cells by employing several signaling pathways, which guarantee that all regions of the genome are duplicated exactly once prior to each cell division. Inaccurate genome duplication, including over- and under-replication and other forms of replication stress, can trigger deleterious consequences such as chromosomal aberrations, stem cell failure, and the development of malignancy[1–4].

As cells emerge from mitosis, chromosomes begin a serial recruitment process resulting in the formation of a protein complex (known as the pre-replication complex, or the replication licensing complex) that includes helicases and accessory proteins essential for genome duplication[5,6]. The loaded helicases are inactive during the first growth (G1) phase of the cell cycle prior to the onset of DNA synthesis. During the synthesis (S) phase, cyclin-dependent kinases activate the helicase and facilitate the recruitment of additional components[5] that allow the helicases to unwind DNA, possibly by forming a phase-separated molecular machine[7]. On each chromosome, replication starts at distinct genomic regions termed replication origins, which initiate replication sequentially during the S phase. Once the helicases incorporated into pre-replication complexes at replication origins are activated, they form active replication forks[5,8] whereas the other pre-replication complex proteins responsible for recruiting helicases to chromatin become inactive or are degraded. This process prevents the further, possibly deleterious recruitment of helicases to the genomic regions already undergoing duplication[9–12].

In metazoans, pre-replication complexes are bound to chromatin in excess, as chromosome duplication initiates only at a fraction of pre-replication complexes bound origins. Although a fraction of replication origins are constitutive, initiating replication in a wide range of cells, other origins exhibit flexible initiation and are activated only within distinct cell types, reflecting preferential association with specific chromatin modifications[13–17]. The choice of replication origins activated during each cell division cycle is also flexible, with some origins initiating frequently while others initiate DNA replication intermittently[15,17]. Excess potential origins that are bound by pre-replication complexes do not initiate replication during normal mitotic growth (dormant origins). These dormant origins can be activated under distinct conditions that slow or stall replication fork elongation (replication stress[2,18–20]), suggesting that the recruitment of pre-replication complexes to intermittent or dormant origins might serve as redundancy to ensure the prevention of incomplete genome duplication. The flexible choice of replication origins could also preserve genome integrity by coordinating replication with transcription and chromatin assembly on the shared chromatin template[14,17].

In normal proliferating cells with intact DNA damage repair pathways, the precise selection of replication initiation sites ensures the completion of genome duplication exactly once prior to each cell division. Replication is often less constrained in cancer cells, which are prone to replication-related errors due to compromised regulatory and checkpoint pathways. For example, partial re-replication frequently occurs in cancer, reflecting the increased abundance of the pre-replication complex components and decreased Geminin levels[12,21] as well as oncogene activation[4]. Partial genome re-replication can also be triggered by the deficiency of RepID, a replicator-specific binding protein[22] that plays a role in ubiquitylation of chromatin substrates[23], or by inhibition of DOT1L, a methyltransferase that catalyzes the demethylation of the replication-origin-associated histone H3 on lysine 79 (H3K79Me2)[24]. Because massive re-replication can drive cell death specifically in checkpoint-compromised cancer cells, both CDT1 stabilization by inactivation of ubiquitin-mediated degradation and inhibition of DOT1L are currently being explored as novel anti-cancer therapeutic strategies[25–31].

Given that genome re-replication is a common avenue to genomic instability and considering its potential as a strategy for chemotherapy, it is important to understand in detail its mechanics and downstream consequences. Here, we report that re-replication progresses slowly, exhibits aberrant replication fork dynamics and a skewed distribution of replication initiation that selectively over-duplicates early-replicating genomic regions. Unlike other instances of replication stress with dormant origin activation, such as when replication slows down as a result of nucleotide depletion or in cells exposed to DNA damaging conditions, the re-replication process preferentially utilizes a subset of the replication origin pool typically used during normal growth.

## Results

**Massive genome re-replication is accompanied by altered replication dynamics.** To determine replication dynamics in cells undergoing re-replication, we first sought to establish an experimental system in which the majority of cells re-initiate DNA synthesis on post-replicated DNA. To that end, we have explored several avenues for triggering re-replication. We have previously shown that re-replication could be triggered by inhibiting Skp2, a key component of CRL1, in a RepID-deficient cell background that prevented the recruitment of the alternative ubiquitin ligase CRL4 to chromatin[23]. As previously reported[25,26,32], we could trigger massive re-replication in HCT116 and U2OS cells by exposure to the NEDDylation inhibitor MLN4924 (pevonedistat), a drug currently tested in clinical trials and known to inhibit both CRL1 and CRL4[25]. For example, as shown in Fig. 1a, cell populations exposed to MLN4924 exhibited a high prevalence of cells with DNA content above G2/M levels that were incorporating the thymidine analog 5-Ethynyl-2′-deoxyuridine (EdU), indicating partial or complete chromatin re-replication.

A detailed analysis of the kinetics and dose-dependence of re-replication, shown in Supplementary Fig. 1, demonstrated that re-replication was detected in cells exposed to 250 nM of MLN4924 as early as 8 h after addition of the drug (7.65% over-replicating >4 N cells in 250 nM MLN4924 treated HCT116 cells versus 2.02% in control cells) while nearly 80% of all cells were undergoing re-replication after 48 h. Exposure of HCT116 cells to a very low dose of 31.25 nM MLN4924 for 48 h triggered partial re-replication whereas higher doses of 125 and 250 nM MLN4924 resulted in re-replication in nearly all the cells (Supplementary Fig. 1b). Similar results were observed in U2OS cells with higher concentrations of MLN4924 (Supplementary Fig. 1c left panels) and when cells were exposed to another NEDDylation inhibitor, TAS4464[31] (Supplementary Fig. 1c, right panels). Because MLN4924 efficiently induced massive re-replication in a dose- and time-dependent manner (Fig. 1a, Supplementary Fig. 1) this drug was selected as the pharmacological re-replication trigger of choice for the remainder of the study.

NEDDylation inhibitors such as MLN4924 inhibit CRL4 and CRL1 ubiquitin ligase complexes that target the replication licensing factor CDT1 for degradation[25]. In accordance with this mechanism, immunoblotting confirmed increased CDT1 levels starting as early as 4 h of MLN4924 treatment in HCT116 cells (Supplementary Fig. 1d). MLN4924 treated cells continued DNA synthesis concomitant with increased cyclin B levels (Supplementary Fig. 1e) and histone H3 did not undergo phosphorylation on Ser10, a hallmark of chromatin condensation (Supplementary Fig. 1f). As shown in Supplementary Fig. 1g, elutriated cells that were released into MLN4924 during the G1 phase of the cell cycle had entered S-phase during the normal time-frame and continued to replicate DNA with DNA content of >4 N. Together, these observations suggest that cells initiate a second round of replication prior to the completion of S-phase,

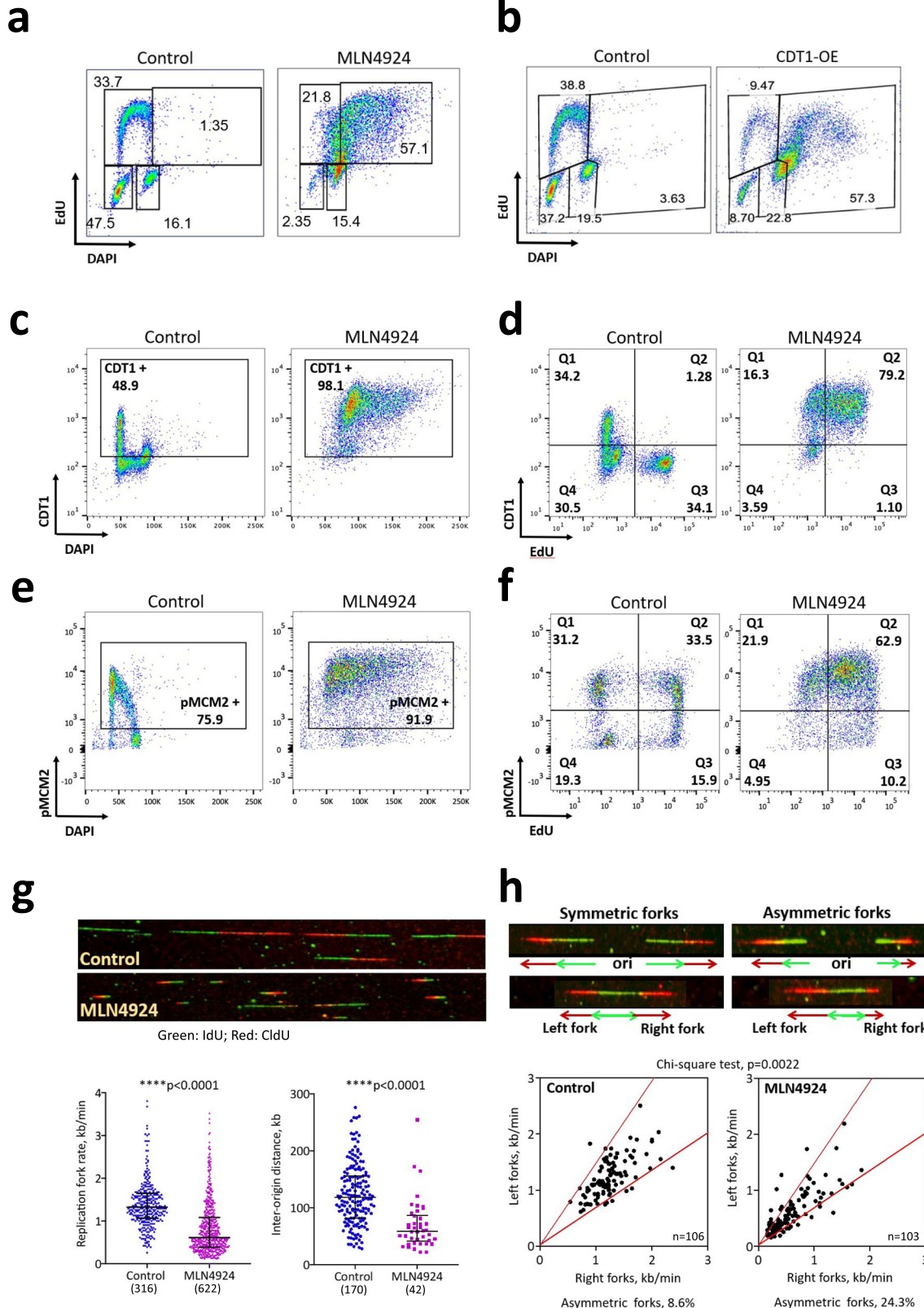

and chromatin condensation and cyclin B degradation, which mark the onset and progression of mitosis, did not occur upon inhibition of CRL4 activity.

As shown in Fig. 1b, we have also observed marked re-replication when we triggered CDT1 overexpression directly (CDT1-OE; these studies were performed in cells expressing a doxycycline-inducible CDT1 to circumvent the lethality of stable overexpression of CDT1[33]). In cells undergoing normal mitotic growth, CDT1 levels were high during the G1 phase of the cell cycle (Fig. 1c, control) but were very low in S phase (Fig. 1c, d, control, EdU-positive cells). CDT1 levels increased, but still remained quite low, during the G2/M phase, consistent with a

**Fig. 1 Massive genome re-replication is accompanied by altered replication dynamics. a** HCT116 cells were treated with 250 nM MLN4924 for 24 h. Cell cycle changes were monitored by flow cytometry, and percentage of cells in each cell cycle stage is shown. See Supplementary Fig. 1 for gating strategy, schematic representation of cell distribution at the different cell cycle stages and additional time points and dose of MLN4924 treatment. Fluorophore Alexa 647 (APC) was used for EdU for all the flow cytometry experiments. Got similar results from 2 or more independent experiments. **b** Doxycycline inducible CDT1-GFP plasmid was stably transfected into U2OS cells (inducible CDT1-U2OS cells). Inducible CDT1-U2OS cells were cultured without (control) or with (CDT1 OE) doxycycline for 24 h (See Supplementary Fig. 3 for 48 h). **c, d** Changes in CDT1 levels (Fluorophore Alexa 488 (FITC) corresponding to cell cycle progression in HCT116 cells treated with 250 nM MLN4924 for 24 h are indicated by DAPI-DNA versus CDT1 (**c**) and by CDT1 versus EdU (**d**). **e, f** Changes in MCM2-pS139 levels (Fluorophore Alexa 488 (FITC) corresponding to cell cycle progression in HCT116 cells treated with 250 nM MLN4924 for 24 h are indicated by DAPI-DNA versus MCM2-pS139 (**e**) and by MCM2-pS139 versus EdU (**f**). **g** Replication profiles in HCT116 cells treated with 250 nM MLN4924 for 30 h measured by DNA combing. Cells were labeled with the thymidine analog IdU for 20 min (green), then CldU (red) for 20 min before collecting. Representative images of DNA combing for both the control and MLN4924-treated samples are shown (top). The replication fork progression rates (bottom left) and replication inter-origin distances (bottom right) are expressed as the median and interquartile range (total number of fibers counted are shown under the sample name). Mann–Whitney test was used for the statistical significance. $p < 0.0001$ by using the two-sided test for both fork rate and origin distance. Similar results were obtained from 3 independent experiments. **h** MLN4924 induced asymmetric replication forks in HCT116 cells. Cells were treated as in Fig. 1g. When the difference between the lengths of left forks and right forks emanating from the same origins was greater than 30%, these forks were classified as asymmetric forks. The percentages of asymmetric forks from each group are shown. The total number of forks analyzed are shown at the bottom right corner. Chi-square test was used for the statistical significance of the percentage difference of asymmetric fork between control and MLN4924 treated samples. Similar results were obtained from 2 more independent experiments.

role of CDT1 in mitotic kinetochores[34,35]. In contrast, CDT1 levels were very high in almost all cells exposed to MLN4924. For example, after exposure to MLN4924 for 24 h, most cells were CDT1-and-EdU-positive (Fig. 1c, d, MLN4924), suggesting that CDT1 accumulation promoted replication re-licensing. We have also observed that MCM2-pS139, a component of the MCM replicative helicase that is phosphorylated by DDK prior to initiation of DNA replication, gradually dissociated from chromatin upon the onset of S-phase in normally replicating cells but remained high on chromatin in EdU-positive, MLN4924-treated cells (Fig. 1e, f).

To characterize replication fork dynamics in re-replicating cells, we sequentially labeled MLN4924-treated and control HCT116 cells with IdU for 20 min, followed by CldU for 20 min. Cells were harvested, and DNA was isolated and subjected to molecular combing to detect IdU (green) and CldU (red)[36]. DNA combing (representative images shown in Fig. 1g, top panel) showed that MLN4924-treated cells exhibited significantly slower replication fork (Fig. 1g, left) and shorter inter-origin distances (Fig. 1g, right), suggesting that during re-replication, slow replication was accompanied by a significant increase in the frequency of replication initiation events. Notably, we have not observed dual-labeled signals (colocalized IdU and CldU)[37] after sequential incorporation of CldU and IdU. As shown in Supplementary Fig. 2, we also did not detect dual-label signals when we prolonged the first labeling by a few hours (incorporation of CldU for 2 or 5 h followed by an IdU pulse) but we could detect frequent re-initiation in MLN4924 treated cells labeled with CldU for 16 h and then pulsed with IdU. These observations suggest that re-replication did not occur immediately after the initial replication event.

We also observed an increased frequency of asymmetric replication forks in MLN4924-treated cells (representative images shown in Fig. 1h, top panel). For example, in the representative experiment shown in Fig. 1h, the prevalence of asymmetric forks was 8.6% in normally replicating cells and 24.3% in MLN4924 treated cells. Replication fork asymmetry was consistent and reproducible, suggesting that replication forks stall at a high frequency in re-replicating cells.

Similar results were observed in CDT1-OE cells, expressing a doxycycline-inducible CDT1 that was tagged with GFP. CDT1-OE cells exhibited increased cell size and granularity (Supplementary Fig. 3a) and demonstrated re-replication after continuous exposure to doxycycline (Supplementary Fig. 3b). Notably, cells that lost CDT1-GFP expression did not undergo re-replication in the

presence of doxycycline (Supplementary Fig. 3c). Similar to MLN4924 treatment, doxycycline-induced CDT1 overexpression was also accompanied by Cyclin B stabilization (Supplementary Fig. 3d), suggesting that stabilization of Cyclin B reflected the absence of mitotic progression upon stabilization of CDT1 and was not a disparate consequence of CRL4 inhibition. Single-fiber analyses demonstrated that CDT1-OE, similar to MLN4924 treatment, triggered slow replication, short inter-origin distances, and asymmetric replication forks (Supplementary Fig. 3e, f).

The consequences of the persistent presence of CDT1 on chromatin include massive DNA damage and the induction of senescence[9,26,37,38]. In concordance, both MLN4924 treated and CDT1-OE cells showed significantly increased levels of the DNA damage and replication stress markers γH2AX, p-RPA, and pChk1S317 (Supplementary Fig. 4a, b). The increase in p-RPA was transient and appeared earlier (Supplementary Fig. 4c) whereas the levels of γH2AX increased continuously, indicating replication stress-associated DNA damage (Supplementary Fig. 4c). Re-replication was also associated with significant increases in ROS levels (Supplementary Fig. 4d) as reported[39]. Consistent with re-replication induced senescence[40], CDT1-OE cells increased in size, showed strong staining for the senescence marker β-galactosidase (Supplementary Fig. 4e).

**Altered genomic distribution of re-replicated DNA.** We then investigated the extent of re-replication on a genome-wide scale. We used a BrdU/CsCl gradient (a variation of the Meselson-Stahl assay[41], outlined in Fig. 2a) to isolate and sequence re-replicated DNA in cells labeled with BrdU for 14 h. This timeframe was expected to enable maximal BrdU substitution in a single DNA strand in normally replicating cells without allowing cells to initiate a second round of DNA replication. The second round of replication would have resulted in BrdU substitution in both DNA strands. As a control, exponentially growing cells were labeled with BrdU for 48 h (resulting in BrdU substitution in both DNA strands). Following the BrdU labeling, genomic DNA was fragmented and fractionated using a CsCl gradient. We detected DNA substituted with BrdU on both strands (heavy-heavy DNA or HH DNA) in cells exposed to MLN4924 (200 nM and 400 nM) for 14 h (Fig. 2b). We then sequenced the DNA from the heavy-heavy fractions in MLN4924 treated samples and in the control sample labeled with BrdU for 48 h. Consistent with recent yeast Re-rep-Seq data[42], we have observed that re-replication was distributed unevenly throughout the genome, with some regions clearly over-represented in re-replicating cells (Fig. 2c). We

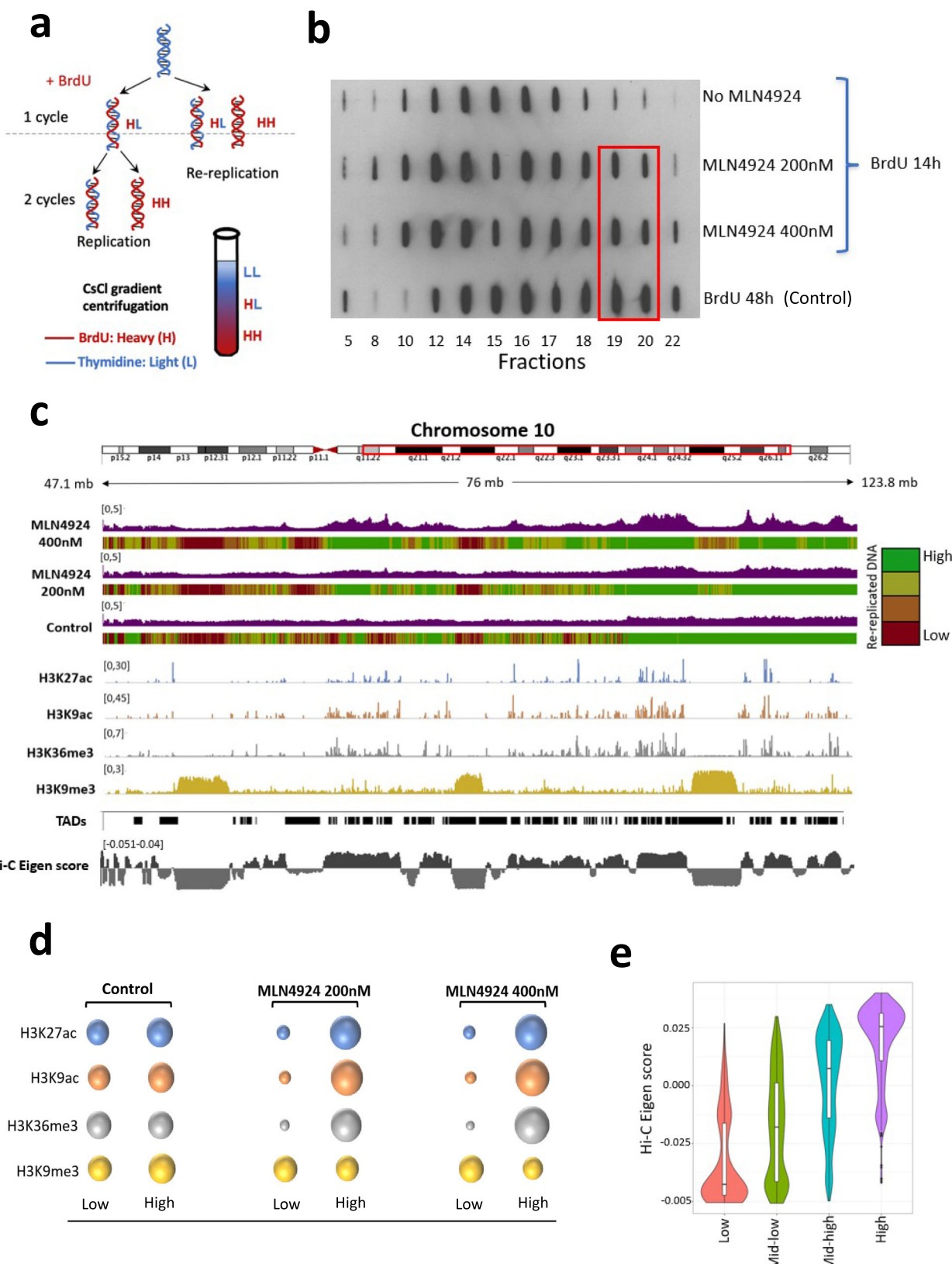

stratified genomic sequences into 4 equal groups based on the extent of re-replication, as reflected in peak heights (high, mid-high, mid-low, and low). As shown in Fig. 2c and Supplementary Fig. 5a, we detected very broad low peaks corresponding to large domains of re-replicated DNA ranging in size between 5 Kb and 5 Mb with a mean size of 58.2 kb for the top quartile of the re-

replicating genome. Consistent with our combing results, we did not detect sharp, high peaks, suggesting that MLN4924 treated cells did not re-initiate multiple rounds of DNA synthesis immediately following the initial round of replication.

We next asked whether regions that were over-represented in re-replicated DNA shared common epigenetic properties. We

**Fig. 2 Skewed distribution of re-replicated DNA. a** Schematic representation of the method used to sequence re-replicating DNA. HCT116 cells were treated with 0, 200, or 400 nM MLN4924 for 14 h. BrdU was added to all samples (including the control) together with MLN4924 to trace replicating DNA. Cells incubated with BrdU for 48 h, which have BrdU incorporated in both strands of DNA, were used as a positive control for re-replication. Genomic DNA was isolated and sonicated to about 0.5–1 kb, and then fractionated by CsCl gradient centrifugation. **b** BrdU levels from equal volumes of selected fractions, from low density (left) to high density (right), were detected using an anti-BrdU antibody. Fractions 19 and 20 from MLN4924-treated and positive control cells (inside the red rectangle), which have BrdU incorporated into both DNA strands, were combined and sequenced. **c** IGV screenshot showing the distribution of DNA re-replication. The sequenced genome was equally divided into 4 groups according to the peak height from green for genomic regions with high peaks (high, mid-high) to red for genomic regions with low peaks (mid-low, low). Tracks for H3K27Ac, H3K9Ac, H3K36Me3, and H3K9Me3 are from the Encode database (see Supplementary Table 1 for track URLs) and are used here to locate euchromatin and heterochromatin domains. TADs and Eigen scores of euchromatin and heterochromatin were calculated using Juicer[74] with the obtained Hi-C data from GEO (GSM2795535)[43]. Positive eigen score indicates euchromatin and negative eigen score is for heterochromatin. The very wide peaks seen after MLN4924 treatment are indicative of open chromatin regions. **d** Bubble chart showing the association of the re-replication most enriched genomic regions (High) and the re-replication least enriched genomic regions (Low) with heterochromatin and euchromatin, respectively. Bubble sizes indicate levels of association. Raw data for the bubble plot are listed in Supplementary Table 2. **e** Eigen scores for the relative euchromatin to heterochromatin levels from low to high abundance of re-replicated DNA are shown. The box range is between 25th percentile and 75th percentile. The whisker length is defined as 1.5 * IQR (Interquartile range). Peaks analyzed from a representative experiment for Low, Mid-low, Mid-high, and High are 285, 605, 753, and 658, respectively. Similar results were obtained from 2 independent experiments.

found that while the sizes of the genomic regions of high peaks and low peaks were similar (Supplementary Fig. 5a), high peaks (regions enriched in re-replicated DNA) were gene-rich (Supplementary Fig. 5b) and highly enriched in euchromatin regions (acetylated histone H3 on lysines 9 and 27, methylated histone H3 on lysine 36), corresponding to active, open, generally early replicating genomic regions (Fig. 2c, d and Supplementary Table 2). Conversely, low peak regions, in which re-replicated DNA was underrepresented, were enriched in heterochromatin marks (methylated histone H3 on lysine 9) that typify condensed, late-replicating chromatin. The distribution of re-replicated DNA did not correlate with the distribution of topologically associated domains (Fig. 2c). However, the high correlation ($R = 0.73$) with Hi–C Eigen scores[43] (Fig. 2c, e and Supplementary Fig. 5c) demonstrated that re-replicated DNA regions were highly enriched in euchromatin.

**Re-replication uses the same set of origins utilized during normal replication.** Given the asymmetric, slow replication observed in re-replicating cells, we asked whether cells undergoing re-replication activate a group of dormant origins similar to those when cells are exposed to agents that perturb or impede DNA synthesis. Although most replication origins are activated at a variable frequency in normally replicating cells, we specifically defined dormant origins as replication initiation sites that are not activated during normal growth and are detected only when cells undergo perturbed replication[20]. To test for dormant origin activation, we developed a method to isolate and sequence replication origins in re-replicating DNA. Following the protocol outlined in Fig. 3a, we first labeled and isolated re-replicated double-stranded DNA (HH—both strands substituted with BrdU) and normally replicated DNA (HL—only a single strand substituted with BrdU) in MLN4924 treated cells by BrdU-CsCl gradient. Then, we followed the protocol for Nascent Strand-Sequencing (NS-seq)[44,45] to isolate and sequence newly replicated short DNA strands in HH and HL DNA. With this combination of BrdU density gradients followed by NS-seq, we sequenced only nascent DNA strands that were labeled with BrdU on both strands, denoting replication origins that initiated re-replication on already-replicated templates. We also sequenced DNA labeled with BrdU on a single strand (HL) from both control (Control-HL) and MLN4924 treated (MLN4924-HL) samples. As shown in the representative IGV screenshots in Fig. 3b, the distribution of NS-Seq peaks during normal replication (Control-HL and MLN4924-HL) was very similar to the distribution of NS-peaks in re-replicated DNA (MLN4924-HH).

To perform a detailed comparison of the frequency of initiation in nascent DNA derived from cells undergoing normal mitotic growth with that of MLN4924-treated, re-replicating cells, we used peak density plots created with BAMScale (https://github.com/ncbi/BAMscale)[46]. This analysis allowed us to determine whether certain groups of replication origins are preferentially utilized during re-replication (Fig. 3c). Density plots represent peak sizes (number of reads/each peak) across the sample pairs. In such plots, similar replication initiation frequencies are reflected in similar peak sizes across the samples, with most peaks distributed along the diagonal (45 degrees dotted black lines in the diagrams shown in Fig. 3c). For example, a comparison the replication initiation frequencies in nascent DNA isolated during normal S-phase (HL DNA) in both untreated cells (control) and MLN4924 treated cells showed that most peaks were distributed along the diagonal, indicating similar peak sizes that correspond to similar initiation frequencies (Fig. 3c, left). By contrast, comparing replication initiation frequencies in nascent DNA isolated from aphidicolin treated cells with nascent DNA isolated from normally replicated cells (Fig. 3c, right) showed a distinct population of replication origins that exhibited a higher read per peak ratio in the aphidicolin sample, corresponding to activated dormant origins. When we compared re-replicated (HH) nascent strands from MLN4924-treated cells with nascent strands that were not re-replicated (HL), we did not observe a distinct population of activated dormant origins (Fig. 3c, middle panel). These observations suggested that although fiber analyses have shown a higher frequency of initiation in re-replicating cells, dormant origins were not overwhelmingly activated.

Similar results were obtained when we used a different strategy to map re-replicating origins. For this, cells were exposed to 250 nM MLN4924 for 30 h or 45 h, timepoints at which nearly all (30 h) or all the replicating cells (45 h) underwent genome re-replication (Supplementary Fig. 6a, left panel). The cells were collected and nascent strand DNA was prepared and sequenced (Supplementary Fig. 6a, right). DNA from cells not exposed to MLN4924 was used as a control. As shown in the representative IGV screenshots in Supplementary Fig. 6b, the distribution of NS-Seq peaks in the control and MLN4924-treated samples was very similar. As shown in the density plots (Supplementary Fig. 6c, left and middle), most re-replicating origins colocalized with the replication origins utilized during normal mitotic growth. Again, this result differed from the observed activation of dormant origins in aphidicolin treated cells and in cells depleted of SIRT1[47], where peaks that did not initiate replication in SIRT1-

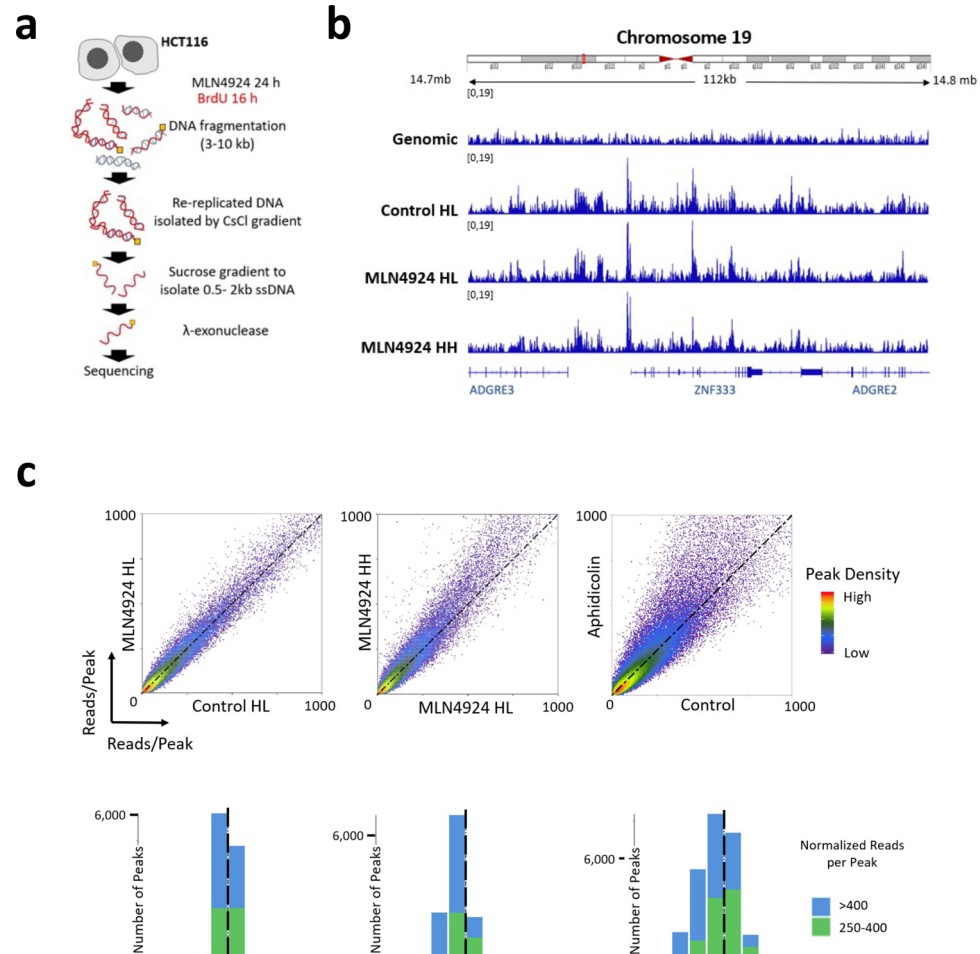

**Fig. 3 Re-replication is driven by the same set of origins utilized during normal mitotic growth. a** Schematic representation of the method used to map replication origins in re-replicating DNA. Cells were treated with MLN4924 for 24 h, including exposure to BrdU for 16 h. Genomic DNA was fragmented and re-replicated DNA (MLN HH DNA from cells incubated with BrdU for less than an entire doubling period) was isolated by BrdU-CsCl gradient. Normal replicated DNA (Control HL DNA and MLN HL DNA, only one DNA strand labeled with BrdU) was also collected to map normal replication origins as a control. Following the CsCl gradient, newly replicated nascent DNA was isolated from both normal- and re-replicated DNA using sucrose gradient and lambda exonuclease. DNA fragments that are resistant to λ-endonuclease digestion (RNA primed, yellow squares) are nascent strands[45]. **b** IGV screenshot showing representative nascent DNA outputs in genomic control, during normal mitotic replication (Control-HL and MLN-HL), and during re-replication (MLN4924-HH). **c** Density plots (top panel) comparing replication origin usage in 2 normal mitotic replication samples (Control-HL and MLN4924-HL, left) and in one normal and one re-replication samples (MLN4924-HL and MLN4924-HH, middle). The location of each data point is proportional to the number of reads per origins. Origins that initiate replication with similar frequency during normal mitotic growth and in re-replicating cells are located on the diagonal dotted line; for the middle panel, origins above the diagonal dotted line initiate more frequently in re-replicating cells and vice versa. The right panel used as positive control for dormant origin activation showing augmented origin activation in aphidicolin treated cells (0.8 μM of Aphidicolin for 24 h). Reads per origin data of the above density plots were further divided into 10 fractions ranging from origins that showed the highest ratio of initiation frequency of y-axis sample (for example MLN4924 HH of the middle panel) vs. x-axis sample (for example MLN4924 HL of the middle panel) to origins that showed the highest ratio of initiation frequency of x-axis sample vs. y-axis sample. The number of peaks in each of the 10 fractions is shown as bar graphs (bottom panel) to show the cumulative distribution of small peaks with 250–400 reads (green) and large peaks >400 reads (blue). The vertical dash line in the middle equal to the diagonal dash line in the above density plot.

proficient cells were clearly observed in SIRT1 depleted cells (Supplementary Fig. 6c, right). As shown in Supplementary Fig. 6d, e, most origins used during normal replication were also used during re-replication induced by over-expression of CDT1 (in U2OS cells harboring doxycycline inducible CDT1). We therefore concluded that re-replicating cells largely utilized the same set of origins as is used during normal mitotic growth, albeit at a higher frequency.

**Early replicating origins are preferred during massive re-replication**. DNA replication follows a stringent spatial order,

forming distinct replication foci patterns in early, mid, and late S phase. As shown in Fig. 4a (and in the accompanying DAPI-stained images shown in Supplementary Fig. 7a), normally growing cells exhibited very typical EdU foci patterns for early (ES), middle (MS), and late (LS) S phase. In contrast, almost all the MLN4924-treated samples were larger and displayed homogeneous replication foci similar to the early-S-phase patterns, with very few exhibiting mid-late-S-phase foci (Supplementary Table 3). These observations suggested that although re-replicating cells utilized the same pool of replication origins as the pool used in normally replicating cells, those re-replicating

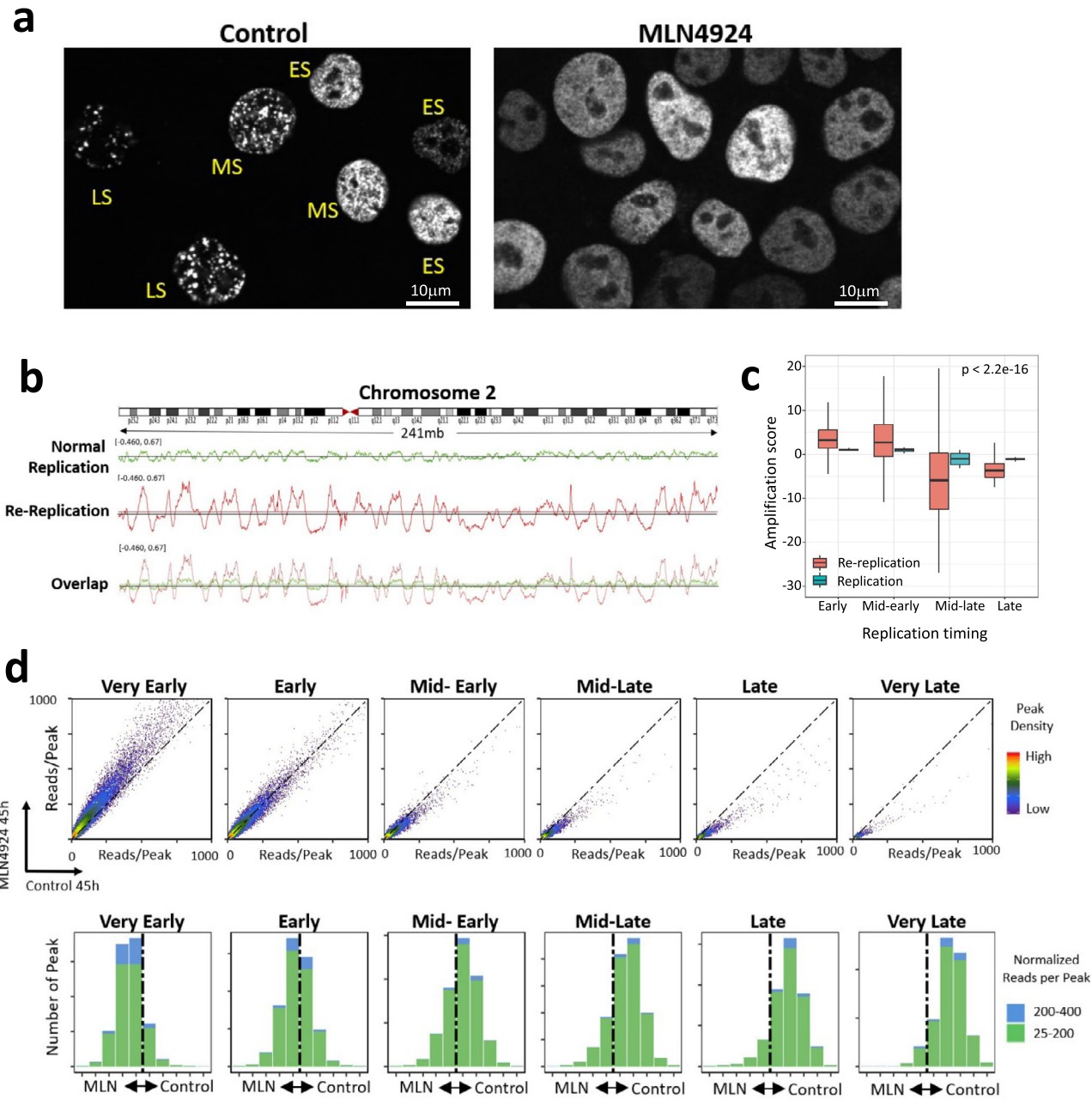

cells lost the spatial distribution characteristic of normal replication.

The observations described above suggested that although early-replicating genomic regions are more prone to re-replication than late replicating regions, the distinction between early and late replication regions is preserved. This preservation is unlike the observed loss of replication timing information in cells undergoing re-replication following geminin depletion, where replication re-initiates at the G2 stage of the cell cycle[33]. To assess if replication timing information is indeed preserved in CDT1-stabilized, MLN4924-treated re-replicating cells and if early-replicating regions are over-represented, we determined the relative replication time of each genomic portion during normal mitotic growth and in re-replicating cells. The order of replication was determined using a variation of the Timex method[48,49] that is based on the assumption that in asynchronously growing cells, early-replicating regions are present in higher copy numbers than late-replicating regions. We measured whole-genome copy number in asynchronous, exponentially growing cells in which more than 50% of the cells were in S phase, as well as in cells in the G1 phase of the cell cycle that were isolated by centrifugal elutriation (Supplementary Fig. 7b). We also measured whole-genome copy number in asynchronous re-replicating cells that were treated with 250 nM MLN4924 for 36 h (89.8% re-replicating cells, Supplementary Fig. 7b). All samples were sequenced with a depth greater than 30×. Log$_2$ values of the copy number in asynchronous replicating and re-replicating cells, normalized to cells in G1, were used to determine the replication timing (Supplementary Fig. 7c).

The screenshot in Fig. 4b shows the distribution of genomic copy number for the entire chromosome 2, with copy number fluctuations shown side-by-side with the same y-axis scale. This analysis suggests that the order of DNA replication (the location of copy number peaks and troughs) was similar in cells undergoing normal mitotic growth and in re-replicating cells, but earlier-replicating genomic regions were over-represented in

**Fig. 4 Replication origins that initiate replication early during normal mitotic growth are over-activated during re-replication. a** HCT116 cells were treated with 250 nM of MLN4924 for 30 h and EdU click-it to label S phase cells. Control cells showed typical S phase cell foci patterns (ES: early S phase; MS: mid-S phase; LS: late S phase). MLN4924 treated cells lost these typical S phase replication foci patterns, showing ES-like foci patterns (see Supplementary Fig. 7a for more details). Three biological repeats yielded similar results. **b** Total genomic DNA of HCT116 cells was purified from G1 cells, asynchronous normal mitotic growing cells and asynchronous re-replicating cells induced by MLN4924 as shown in Supplementary Fig. 7b and sequenced at >30× coverage. IGV screenshot of replication timing of chromosome 2 for normal replication and re-replication (same y-axis scale) showing increased copy number at the early replication genomic regions and decreased copy number at the late replication genomic regions in re-replicating cells. **c** Genomic regions were divided into four equal groups according to their replication timing and amplification scores of these segments were calculated in control and re-replicating nascent DNA. 5051, 8624, 6746 and 3076 origins from one representative experiment for early, mid-early, mid-late and late, respectively, were analyzed. Box plots are showing the median, 25th and 75th percentiles as box boundaries and the 5th and 95th percentile ranges. Wilcoxon test was used for the statistical significance analysis. $p < 2.2e-16$ by using the default two-sided test. 3 independent experiments showed similar results. **d** Top panel: Density plots comparing reads per peak (reflecting the frequency of initiation per each origin) in groups of replication origins from cells undergoing normal replication (control) and re-replication (MLN4924 for 45 h). Origins were classified into six replication timing groups, ranging from very early to very late, using the replication timing data illustrated in panel **b**. Data points from origins that exhibited similar frequencies of initiation during normal replication and re-replication are expected to localize on the diagonal dotted line. Data points above the diagonal dotted line represent origins that initiated more frequently during re-replication than during normal S-phase, and vice versa. *Bottom panel*: origins in the density plot were further divided into 10 fractions ranging from origins that showed the highest ratio of initiation frequency during re-replication *vs.* during normal replication (MLN 45 h, left side of each bar graph) to origins that showed the highest ratio of initiation frequency during normal replication vs. during re-replication (Control 45 h, right side of each bar graph). The number of peaks in each of the 10 fractions showed the gradually shift from more origins initiated in MLN4924 treated sample than in control at early replicating genomic regions to fewer origins initiated in MLNN4924 treated sample than in control at late replication genomic regions.

re-replicating cells whereas late-replicating genomic region were under-represented. To extend this analysis to the entire genome, we stratified all genomic regions into four groups based on replication order and calculated the extent of over- or under-representation of these segments in control and re-replicating nascent DNA. As shown in Fig. 4c, although early replicating regions were expected to exhibit some over-representation in nascent strands from asynchronously proliferating cells, nascent strands representing early replicated regions from re-replicating cells showed a markedly enhanced level of enrichment (the mean enrichment was 2–5 fold higher than the enrichment observed in normal replication). These observations suggest that early-replicated genomic regions preferentially undergo re-replication to an extent that could not be attributed solely to the over-abundance of early replicating regions in asynchronously proliferating cells, possibly undergoing multiple rounds of replication as previously shown in Xenopus egg extracts[38]. This enrichment in early-replicating origins was consistent with the observation that re-replication preferred open chromatin regions and showed early-S-phase-like re-replication foci patterns.

To test directly whether the over-represented, earlier-replicating genomic regions in re-replicating cells reflected re-initiation at origins located within the early-replicating portion of the genome, we directly compared the frequency of replication initiation events at origins replicated at different stages of S phase in untreated and MLN4924-treated samples. All origins were stratified into six groups according to replication order, ranging from very early to very late. Then, we used BAMScale to create density plots measuring the number of reads per replication origin peak during normal mitotic growth and during re-replication. As shown in Fig. 4d, very early origin peaks were mapped above the diagonal, suggesting that early replication origins initiate replication more frequently in re-replicating cells than in cells undergoing normal mitotic growth. This over-initiation was not evident in later-replicating genomic regions; late-replicating origins initiated more frequently in cells undergoing normal mitotic growth than in re-replicating cells. Similar results were also obtained when we used a second experimental strategy to isolate re-replicating nascent strands (Supplementary Fig. 8a). As a control, we have not detected initiation timing shifts when we compared 2 sets of normal replication samples (Supplementary Fig. 8b). Altogether, these results show that re-

replication occurs throughout the genome with a prevalence towards open chromatin and early replicating regions.

**The frequency of re-replication reflects the distribution of pre-replication complexes.** To elucidate the molecular mechanism underlying the preferred activation of origins in early-replicating regions, we asked if this preference reflected a skewed distribution of CDT1 binding sites. To that end, we performed Chromatin Immunoprecipitation followed by Sequencing (ChIP-Seq) with antibodies directed against CDT1 using HCT116 cells synchronized by double thymidine in G1 as well as middle S-phase (MS). ChIP-seq data were processed against input. As shown in Supplementary Fig. 9a, CDT1 chromatin binding was primarily detected in G1 cells, consistent with the reported degradation of CDT1 at the onset of S-phase. Figure 5a and Supplementary Fig. 9b show representative IGV screenshots of CDT1 ChIP-Seq data obtained from normal asynchronously proliferating cells (representing primarily interactions at the G1 phase) and re-replicating cells (in which CDT1 remains on chromatin during re-replication). Representative images from chromosome 19 and chromosome 4 are shown, demonstrating increased CDT1 chromatin binding on early replicating DNA and diminished CDT1 binding on late-replicating regions. This selective enrichment was markedly enhanced in re-replicating cells vs. in control, normally proliferating cells. Figure 5b shows whole-genome analyses of CDT1 ChIP-seq data stratified into 6 groups based on replication order (from very early to very late replicating regions) measuring the distribution of CDT1 binding sites colocalized with replication origins. As shown, CDT1 was enriched at replication origins, and this colocalization increased in early-replicating genomic regions in re-replicating cells.

We further asked if the distribution of CDT1 binding sites reflected diminishes binding in heterochromatin regions that replicated late in re-replicating cells. To that end, we calculated the association of normalized CDT1 binding scores (expressed as FPKM) around CDT1-bound origins stratified in accordance with replication timing. As a control, we also determined the origin-associated CDT1 signal distribution in random genomic regions (Supplementary Fig. 9c), which showed low levels of CDT1 binding in both normal and re-replicating cells. The violin plots in Fig. 5c show that the binding of CDT1 at origins was skewed towards early replicating regions in normally growing cells, and

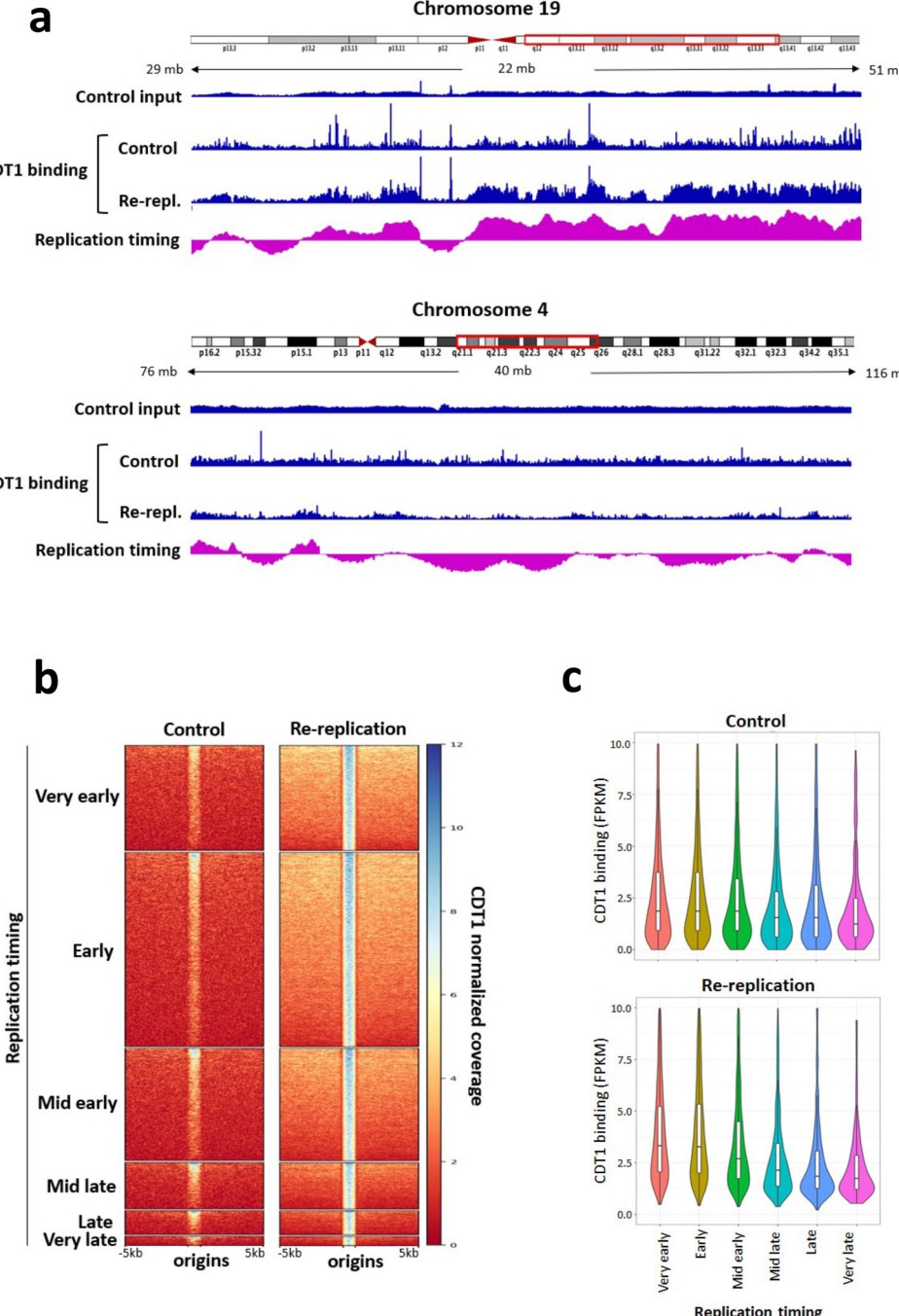

**Fig. 5 CDT1 is enriched at early origins.** CDT1 ChIP-seq was performed in HCT116 cells exposed to MLN4924 for 36 h (re-replication). **a** IGV screenshots of the distribution of CDT1 binding sites on parts of chromosome 4 and 19 in normally proliferating (control) and re-replicating cells. Input from control cells is shown at the top (for screenshot with input for both control and re-replication, see Supplementary Fig. 9). DNA replication timings of the genomic regions are shown at the bottom. **b** Heatmap summarizing the normalized sequencing coverage of CDT1 ChIP-seq in normally replicating control and re-replicating samples. CDT1 peaks called against input were first intersected with origins (CDT1 binding sites overlapping with origins), then were divided into 6 groups based on the replication timing data in Fig. 4. **c** Enrichment of CDT1 centered on replication origins residing in replication timing domains stratified as in **b**, was shown as violin plots. The box range is between 25th percentile and 75th percentile. The whisker length is defined as 1.5 * IQR (Interquartile range). Two biological repeats yielded similar results.

that this preferential binding was amplified in re-replicating cells (for CDT1 peak distribution at different replication timing fractions and statistical analysis, see Supplementary Tables 4 and 5).

The observations described above are consistent with the hypothesis that although DNA re-replication is accompanied by slow DNA synthesis and increased frequency of initiation, the

choice of re-replicating origins reflects the abundance of pre-replication complexes with chromatin at the G1 phase prior to the onset of re-replication[50]. To test directly for the locations of pre-replication complexes on chromatin, we asked whether MCM2-pS139 binding sites colocalize with replication origins during normal growth and during re-replication. As shown in Supplementary Fig. 10a–d, MCM2-pS139 binding sites enriched at

replication origins, and this enrichment dramatically increases and is skewed towards early replicating DNA in re-replicated DNA. These observations further suggest that the locations of pre-replication complexes during replication licensing dictate the distribution of replication initiation events during both normal replication and re-replication.

## Discussion

The results reported here characterize replication re-initiation in cancer cells containing persistent pre-replication complexes (pre-RCs) on chromatin during the S-phase of the cell cycle. Unlike normal mitotically growing cells, which activate a fraction of their potential replication origins on each chromosome each cell division, resulting in inter-origin spacing of 100–150 kb, we found that re-replicating cells show a higher frequency of initiation from origins positioned in closer proximity to one another. Surprisingly, these additional origins are derived from the same pool of potential replication initiation sites as the origins used during normal mitotic proliferation. We also observed that DNA synthesis during re-replication is slower than during normal mitotic growth, and that re-replication is accompanied by a high incidence of replication fork asymmetry reflecting the frequent stalling of replication forks that can lead to DNA damage, ROS production and eventually trigger senescence. Our results suggest that re-replication begins during the mid-to-late S-phase, prior to the completion of the first round of replication, and that genomic regions that associate with the euchromatic, early-replicating portion of chromatin are preferentially re-replicated.

As illustrated in Fig. 6, replication origins in re-replicating cells consist of sequences that initiate replication at various frequencies during normal growth (frequent origins and intermittent origins). Re-replication occurs unevenly, primarily increasing the frequency of initiation from intermittent origins located in the early-replicating portion of the genome but occurs at a low frequency in the late-replicating portion of the genome, similar to the pattern observed in yeast and in simulated re-replication in mammalian cells[42]. As suggested by our copy number analyses, the earliest-replicating regions can undergo multiple initiations in re-replicating cells, as previously shown in Xenopus[38], whereas the late-replicating regions are depleted. This pattern might reflect epigenetic modifications, as early-replicating origins are associated with euchromatin and therefore might represent a population of readily accessible licensed origins with a high density of closely spaced pre-replication complexes[14,15,50–52].

The preferential duplication of early-replicating regions mirrors the unequal distribution of CDT1 on pre-replicated chromatin. During normal mitotic growth, CDT1 recruits the MCM helicases to facilitate initiation of DNA synthesis, which is followed by prompt removal of pre-replication from chromatin via the ubiquitination and targeted degradation of CDT1[1,9,12,35,53]. In re-replicating cells, CDT1 is not degraded and continues to recruit pre-replication complexes, resulting in the re-initiation from the same pool of origins as those used during normal mitotic growth, albeit at a higher frequency. The unequal distribution of CDT1 on pre-replicated chromatin, already evident during the G1 phase, ensues an unequal re-recruitment of MCM helicases to already-replicated potion of chromatin in re-replicating cells[50,54].

Our analyses demonstrate that the consistent, tissue-specific replication order (replication timing) evident during normal growth is preserved during genome re-replication, but we find that cells do not wait to complete the first round of DNA replication prior to the activation of the second set of initiation events. Rather, before completing replication of the entire genome, cells initiate replication at regions that were previously duplicated. This pattern differs from the replication patterns observed for other known instances of genome re-replication during embryonic development[55], when cells undergo numerous iterations of entire genome duplication, with S phases and M phases alternating to re-license origins anew after the completion of each replication cycle. Because re-replicated DNA can be more susceptible to damage and breakage, the preferential re-replication of early-replicated DNA can be linked to the observed clustering of DNA breaks in euchromatic regions[56,57].

Re-replication is observed in certain developmental systems as a mechanism for controlling the development of specialized cellular

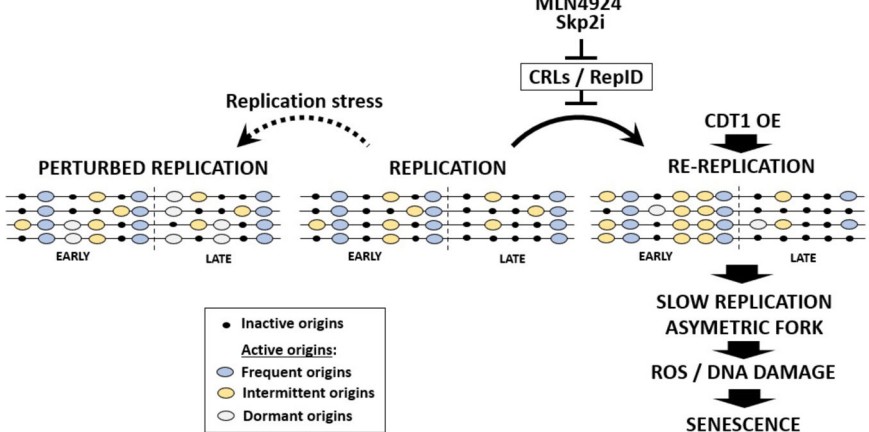

**Fig. 6 A model of replication origin usage during re-replication.** Re-replication can be induced by pharmacological inhibition of the CRL ubiquitin ligases complexes (MLN4924, Skp2 inhibitor) or by depletion of cellular components recruiting CRLs to chromatin (e.g., RepID). Both strategies lead to CDT1 stabilization and abnormal high cellular levels of CDT1. Re-replication can also be induced by CDT1 overexpression. Re-replicating origins are markedly enriched for sequences that are normally replicated during the early stages of genome duplication, and replication initiation is repressed in late-replicating regions. Intermittent origins, defined as origins that are activated intermittently during normal genome duplication, initiate replication at a higher frequency during re-replication. Activation of intermittent origins results in shorter inter-origin distances and slower replication fork progression than during normal mitotic growth, reflecting frequent replication fork stalling. In cells undergoing replication stress, excess initiation triggers activation of dormant origins, which do not initiate replication during normal growth. Re-replication may activate some dormant origins, but at a notably lower frequency compared to dormant origins induced by aphidicolin and SIRT1 depletion. Over-activation of origins and frequent stalling can lead to DNA damage, ROS production and eventually trigger senescence.

functions[12,55], and re-replication can lead to recombination events that shape the evolution of the yeast genome[58]. In normal mammalian somatic cells, re-replication is often fatal, activating a checkpoint control that initiate cell death to avoid carcinogenesis[21,59,60], and is prevented by CDT1 ubiquitination[9,29,53,61,62] or via the interaction of CDT1 with geminin[12,35,53,59,63]. Cancer cells, which often lack the re-replication induced cell death checkpoint, can tolerate limited re-replication, and interfering with the degradation and inactivation of CDT1 by geminin depletion[63] or by inhibiting the activity of cullin-anchored ubiquitin ligases. Notably, re-replication patterns induced by geminin depletion differ from the patterns we observed, as geminin-induced re-replication is triggered in the G2 phase of the cell cycle and is associated with the loss of replication timing information[33] whereas CDT1 stabilization during S-phase by inhibition of CRL4 (as shown here or by Emi1 depletion[64]) begins during the mid-to-late S-phase, prior to the completion of the first round of replication.

Cancer cells can activate dormant replication origins when exposed to agents that slow or halt DNA replication. Because cells that undergo partial genome re-replication also exhibit slow DNA synthesis and undergo DNA damage, we cannot unequivocally exclude the possibility that the slow replication is the cause, and not the consequence, of the activation of additional origins. We also cannot exclude possible limited activation (below our detection threshold) of dormant origins as re-replication progresses. However, we observed that nearly all replication origins activated in cells undergoing re-replication paralleled the replication origins activated during normal S-phase. These observations suggest that the high frequency of initiation in re-replicating cells is derived from frequent activation of intermittent origins on already-replicated chromatin, a mechanism that markedly differs from the dormant origins activated due to other triggers, such as exposure to DNA damaging agents, low nucleotide pools, or depletion of SIRT1[18,20,32,47,65,66].

The existence of mechanisms that normally select a small group of active replication initiation sites from a pool of potential replication origins can facilitate genomic stability, allowing for the complete duplication of the genome when replication stalls. Under such circumstances, excess origin activation prevents under-replication, which can lead to cell cycle perturbations, chromosomal translocations, and DNA breakage in regions with low origin density[2,18,47,67,68]. However, excess initiation of DNA replication can have deleterious consequences, including oncogenic transformation of normal cells and increased genomic instability in cancer[21,26,59,63], consistent with our observations suggesting that massive over-replication can lead to senescence. Those consequences can be exploited therapeutically to induce selective killing of cancer cells[26,29–31,63]. The results reported here imply that circumventing the strong inhibitory interactions that normally prevent excess DNA synthesis can occur via at least two pathways, each activating a distinct set of replication origins. Understanding the interactions underlying both pathways could clarify the mechanisms that monitor and regulate the progression of genome duplication and lead to improved selectivity in targeting cancer.

## Methods

**Cell culture, drugs, and establishment of a CDT1 stable cell line**. Human HCT1116 cells and U2OS cells were grown at 37 °C in a 5% $CO_2$ atmosphere in Dulbecco's modified Eagle medium (ThemoFisher, 10564011), supplemented with 10% fetal calf serum. MLN4924 was purchased from Cayman Chemicals (15217-1). CDT1 cDNA was obtained by reverse transcription and PCR with primers CDT1-F and CDT1-R (Supplementary Table 6) from HCT116 cells. CDT1 cDNA amplified by HindIII_Cdt1F and XhoI_Cdt1R primers (Supplementary Table 6) was inserted to pCMV-Tag2A (https://www.addgene.org/vector-database/6193/). CDT1 sequence was verified using sanger sequencing. CDT1 cDNA was further cloned into the Tet-On 3 G inducible expression system (TaKaRa, 631168) with a flag tag at the N

terminal and an EGFP tag at the C terminal by the In-Fusion HD Cloning system (TaKaRa, 638909) with infusion primers CDT1F and CDT1R (Supplementary Table 6). After transfection of the U2OS cells with the pCMV-Tet3G Vector, stable clones were further transfected with the pTRE3G Vector containing flag-CDT1-GFP. Stable clones were tested for GFP positivity (CDT1-GFP) and re-replication by flow cytometry and presence of fusion protein by western blotting with anti-CDT1 antibody.

**Flow cytometry**. Cells were pulse-labeled with 10 μM EdU for 30–45 min before harvest. EdU staining was performed using the Click-iT EdU kit (ThermoFisher, C10634 or C10633) according to the manufacturer's protocol. For immunodetections with Phospho-MCM2 (Ser139) (MCM2-pS139, Cell Signaling, 12958), and anti-CDT1 (Cell Signaling, 8064), cells were incubated on ice for 10 min in a cytosol extraction buffer (10μM HEPES, pH7.9; 10 μM KCl; 1 μM EGTA; 0.25% NP40; 1× protease inhibitor cocktail and phosphatase inhibitor cocktail), then centrifuged at $800 \times g$ for 5 min at 4 °C. The cell pellets were washed once with a cytosol extraction buffer, then cells were fixed with 4% PFA. EdU click were followed by CDT1 (dilution 1:100) or MCM2-pS139 (dilution 1:100) at 4 °C overnight prior to flow analysis to visualize changes in protein levels during the cell cycle. For the fluorophores, EdU using Alexa 647, MCM2-pS139 (dilution 1:100), and CDT1 using Alexa 488 conjugated anti-rabbit IgG (Thermo Fisher Scientific, A11008) (dilution 1:100). DAPI was used for DNA staining. A BD LSR Fortessa cell analyzer with FACSDiva software and/or FlowJo10.6 were used for cell cycle analysis.

**Microscopy**. Cells were incubated with or without EdU (10 μM) for 30 min, fixed with 4% paraformaldehyde for 10–15 min at room temperature. Cells were then permeabilized with 0.5% Triton X-100 in PBS for 30 min, blocked with 5% BSA for 30 min, and followed by Click-iT EdU labeling with/without antibody staining. EdU labeling by the Click-iT EdU kit (ThermoFisher, C10634 or C10633) was performed according to the manufacturer's protocol. Samples were incubated for 2 h with primary antibodies anti–γH2AX (Millipore, 05-636) and anti-p-RPA (Bethyl labs, A300-245A) at 1:500 dilution followed by incubation for 1 h (dilution 1:500) with secondary antibodies (Alexa 488 conjugated anti-mouse IgG and Alexa 568 conjugated anti-rabbit IgG (Thermo Fisher Scientific, A11029 and A21428)). DNA was counterstained with DAPI. The Zeiss LSM710 confocal microscope and Nikon SoRa super-resolution spinning disk microscope.

**DNA replication analysis by molecular combing**. Analysis of DNA replication by molecular combing was performed, as previously described[36]. First, asynchronous cells were sequentially labeled with 20 μM IdU for 20 min and 50 μM CldU for 20 min, then chased with 200 μM thymidine for 60–90 min. To preserve long genomic DNA fibers, harvested cells were embedded in low melting point agarose plugs. The plugs were incubated in cell lysis buffer with proteinase K at 50 °C for 16 h, washed 3 times with TE buffer, and then melted in 0.1 M MES (pH 6.5) at 70 °C for 20 min. Agarose was subsequently degraded by adding 2 μl of β-agarase (Biolabs). To stretch DNA fibers, DNA solutions were poured into a Teflon reservoir and DNA was combed onto salinized coverslips (Genomic Vision, cov-002-RUO) using an in-house combing machine. Coverslips were visually examined for DNA density and fiber length by YOYO1 DNA staining (Invitrogen). Combed DNA on coverslips was then baked at 60 °C for 2 h and denatured in 0.5 N NaOH for 20 min. Coverslips were blocked for 10 min in 5% BSA/PBS. IdU, CldU, and single-strand DNA were detected using a mouse antibody directed against BrdU (IgG1, Becton Dickinson, 347580, 1:25 dilution), a rat antibody directed against BrdU (Accurate chemical, OBT0030, 1:200 dilution) and a mouse antibody directed against single-stranded DNA (ssDNA) (IgG 2a, Millipore, MAB3034, 1:100), respectively. Incubation with primary and secondary antibodies were performed at room temperature in 1% BSA in PBS for 1 h and 45 min respectively. The secondary antibodies used were goat anti-mouse Cy3 (Abcam ab6946, 1:100 dilution), goat anti-rat Cy5 (Abcam, ab6565, 1:100 dilution) and goat anti-mouse BV480 (Jackson ImmunoResearch, 115-685-166, 1:50 dilution) for ssDNA. Slides were scanned with a FiberVision Automated Scanner (Genomic Vision). Replication signals on single DNA fibers were analyzed using FiberStudio (Genomic Vision). Only replication signals from high-quality ssDNA (not those from DNA bundles nor those located at the end of a strand) were selected for analyses. Experiments were performed at least in duplicate using independent biological isolations of DNA fibers for each experimental condition. The statistical analyses were performed using Prism 9 (GraphPad software) and the non-parametric Mann–Whitney rank sum test.

**BrdU-CsCl gradient to isolate re-replicated DNA**. HCT116 cells were cultured with 50 μM BrdU for the indicated times. Genomic DNA was purified and sonicated to 500–2000 bp. Sonicated genomic DNA from cells cultured with/without BrdU for 48 h were used as BrdU positive and negative control, respectively. In total 300 μg of sonicated DNA were fractionated at $132,000 \times g$ in a Ti75 rotor (Beckman) for 66 h using 6 ml CsCl (1 g/ml in TE). Fractions of 250 μl were collected and the refractory index was measured to confirm the formation of the CsCl gradient. Samples of 20 μl from each fraction were loaded to a positively charged nylon membrane using a Slot Blot Filtration Manifold (PR648, GE

Healthcare Life Sciences, PR648). The presence of BrdU on the nylon membrane was detected with an anti-BrdU antibody (IgG1, Becton Dickinson, 347580, 1:100 dilution). DNA in which both strands had undergone BrdU incorporation was collected and sequenced using the Illumina genome analyzer II.

### Nascent strand DNA sequencing (NS-seq) for re-replicating cells

*Strategy 1: NS-seq in re-replicating cells.* HCT116 cells were treated with 250 nM of MLN4924 for the indicated time in order to have most or all cells in re-replicating cycle. Cells incubated without MLN4924 were used as control. Genomic DNA was purified by phenol/chloroform and ethanol precipitation and used to isolate nascent strands. To isolate nascent strands, DNA was denatured by boiling for 10 min, immediately cooled on ice, and fractionated on a neutral 5–30% sucrose gradient. Gradients were centrifuged at $50,000 \times g$ for 18 h with an SW40 swing bucket rotor. Fragments, 0.5–2 kb, (containing nascent strand DNA and broken genomic DNA) were collected and treated with λ exonuclease to remove non-RNA-primed broken genomic DNA. The remaining single-stranded nascent strand DNA was converted to double-strand DNA using the BioPrime DNA Labeling System (ThermoFisher, 18094011). Double-stranded nascent DNA (1 μg) was sequenced using the Illumina genome analyzer II (Solexa). Sheared genomic DNA was also sequenced to be used for peak calling.

*Strategy 2: isolating re-replicated DNA followed by NS-seq.* HCT116 cells were treated with MLN4924 for 24 h. BrdU was added to cells 8 h after MLN4924 treatment for a total of 16 h of BrdU incorporation, which was less than one doubling time. DNA was purified from these cells and sonicated to 3–10 kb. Re-replicated DNA (in which both DNA strands had undergone BrdU incorporation) was isolated using a BrdU-CsCl gradient. Re-replicated DNA and some DNA that had not undergone re-replication (only one strand having incorporated BrdU) were collected for NS-seq as described below.

### Replication timing in normal and re-replicating cells

HCT116 were treated with MLN4924 for the indicated times and doses. Untreated G1 cells were isolated by elutriation at $750 \times g$, at 4 °C, at flow rate 15 ml/min. DNA from both G1 ( > 98% in G1 phase) and exponential growing (>50% cells in S phase) untreated cells as well as from re-replicating cells were extracted using the Qiagen DNeasy blood and tissue kit (cat# 69581). DNA samples were sequenced on HiSeq using Illumina TruSeq Nano DNA library preparation and paired-end sequencing. The mean coverage was at least 30X depth.

### ChIP-seq

HCT116 cells were treated with 250mN of MLN4924 (re-replication) or without MLN4924 (control, asynchronous cells) for 36 h. Control and re-replicating cells were crosslinked with 1% formaldehyde for 10 min at RT. After adding glycine to quench formaldehyde and PBS wash, cells were resuspended with cytoplasmic extraction with 0.25% NP40 buffer (5 times pellet size, or 500 μl whichever is higher) plus proteinase and phosphatase inhibitor (1×), and incubated on ice for 5 min to remove cytoplasmic proteins. Nuclei pellets were resuspended in 500 μl NP40 buffer, and were sonicate for 65 pulses with 40% output. The supernatants (from about 2 × 10^6 cells) precleared with protein A beads were incubated with 3 μl of CDT1 or MCM2-pS139 antibodies, 80 μl of protein A beads overnight at 4 °C. protein A beads were washed twice with low salt buffer, high salt buffer, lithium chloride buffer, and TE (each spin at $800 \times g$ for 1–2 min), respectively. Samples were eluted, incubated at 65 °C for overnight for the reverse-crosslinking, then purified by Monarch PCR & DNA Cleanup Kit (NEBT1030S). In total 10 ng of ChIPed samples and input (to be used as genomic DNA control to call peaks in ChIPed samples) were used to generate the library. The kits used for the library were NEBNext Ultra II DNA Library Prep Kit for Illumina (NEB, E7805S) and NEBNext Multiplex Oligos for Illumina (Index Primers Set 1 and Set 2)(NEB, E7335S, and E7500S). Sequencing was done by Illumina NextSeq 75 cycle High Output kit.

### Bioinformatic analyses

Raw FASTQ sequencing files were first trimmed with the Trimmomatic (version 0.36)[69] and Trim Galore (version 0.4.5)[69] programs to remove low quality reads. Trimmed FASTQ files were then checked for quality using FastQC (version 0.11.5) [https://www.bioinformatics.babraham.ac.uk/projects/fastqc/]. Trimmed reads were aligned to the hg19 genome using the BWA aligner (version 0.7.17)[70]. Peaks with high read coverages were identified by the narrow MACS2 (version 2.1.1.20160309)[71] peak calling method using genomic DNA sequencing as controls. Peaks were filtered using the peak-score MACS2 metric in R (version 3.5.1) by accepting regions above the inflection-point threshold of peak-scores from the raw output. In order to compare samples by coverage, the BAMscale cov method was prepared with merged nascent-strand regions and alignment files for each sample. Regions were assigned normalized coverage values based on the library size normalization method of BAMscale. Peak density plots comparing sample pairs were created using R, the code is available at the BAMscale GitHub page (https://github.com/ncbi/BAMscale/wiki). For viewing in the genome browser, the BAMscale scale method was used to develop scaled bigwig coverage tracks for each alignment file in the set. Peaks were also compared to the ENCODE database (Supplementary Table 1) for analyses of histone modifications using the GIGGLE[72] search engine. Post-calculation analyses included the development of an inclusion ratio:

$$IncRatio = \frac{\text{Overlaps Between Sample and ENCODE File}}{(\text{Size of ENCODE File} + \text{Size of Sample File})}$$

Subsequent visualizations were prepared using R-scripts, Excel, as well as the Deeptools computeMatrix and plotHeatmap tools[73].

For BrdU-CsCl assay to detect the distribution of re-replicated DNA, samples were analyzed using the bigwig segmentation R-script available on the BAMscale GitHub page. Coverage files were separated into quartiles based on the abundance of re-replicating DNA and visualized using IGV (Fig. 4c). The chromatin features of these re-replication peaks and their surrounding genomic regions up to 35 kb were analyzed. H3K27Ac, H3K9Ac, H3K36Me3, and H3K9Me3 are from the Encode database (see Supplementary Table 1 for track URLs). Hi-C contact matrix for HCT-116 cell line was obtained from GEO (GSM2795535)[43] in.hic format. Topologically associated domains (TADs) and Eigen score of hetero- and euchromatin were calculated using Juicer[74] using the obtained Hi-C matrix.

Replication timing data was processed by the sequencing facility using the *DRAGEN* analysis pipeline (01.003.044.02.05.01.40152). Data received from the facility was then transformed into $\log_2$ ratio coverage tracks using the BAMscale scale method and associated operation $\log_2$ flag. Separation of replication timing regions from very early to very late was completed using the $\log_2$ ratio of re-replication versus G1 (percentile of total ratio-range: 0–10% for Very Early; 10–30% for Early; 30–50% for Mid-Early; 50–70% for mid-Late; 70–90% for late and 90–100% for Very Late, Fig. 6d). R-scripts which were utilized for this task are available in the BAMscale GitHub page.

R ggplot geomboxplot (3.3.3) function was used to create violin plot for the sequencing data. The box range is between the lower quartile (25th percentile, denoted as $Q_1$) to upper quartile (75th percentile, denoted as $Q_3$). The whisker length is defined as 1.5 * IQR (Interquartile range), where IQR is the distance between $Q_3$ and $Q_1$.

**Reporting summary**. Further information on research design is available in the Nature Research Reporting Summary linked to this article.

## Data availability

All the sequencing data were deposited in GEO (GSE172417). Source data are provided with this paper: Histone modification ChIP-seq data from ENCODE database and Hi-C data from GEO (GSM2795535). The source data underlying Figs. 1g, h; 2e; 4c and 5c and Supplementary Figs. 3e, f; 4a; 5a, c; 9c and 10c, d, as well as uncropped blots are provided as Source Data files. All data within the manuscript are available from the authors upon request. Source data are provided with this paper.

## Code availability

Peak density plots comparing sample pairs were created using R, the code is available at the BAMscale GitHub page (https://github.com/ncbi/BAMscale/wiki) (https://doi.org/10.1186/s13072-020-00343-x).

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

## Acknowledgements

This study was supported by the Intramural Research Program of the NIH, Center for Cancer Research, National Cancer Institute (project ZIA BC010411 to MIA). We thank Dr. Shuichi Okubo from Taiho corporation for TAS4464 and for helpful discussions. We thank the CCR core sequencing facility headed by Bao Tran and Jyoti Shetty for expert help with DNA sequencing. We are grateful to Dr. Michael Kruhlak, Dr. Langston Lim, and Dr. Andy Tran (Confocal Microscopy Core Facility, CCR, NCI, NIH) for expert technical assistance in confocal microscopy. We thank the CCR Genomics Core led by Liz Conner for the expert help with ChIP-sequencing, especially to Madeline Wong for her quick and excellent work.

## Author contributions

H.F. and M.I.A. designed the study. H.F., C.E.R., B.L.T., K.U., and M.I.A. designed the experiments. H.F., C.E.R., B.L.T., K.U., R.S., S.M.J., A.B.M., S.Z.Z., S.B.L., M.R., S.T.M., A.M.B. performed the experiments. H.F., C.E.R., B.L.T, K.U., L.S.P., and M.I.A. analyzed the data. J.M.G., S.M., L.S.P. performed sequencing data processing and analysis. H.F., C.E.R., B.L.T, L.S.P., and M.I.A. wrote the manuscript.

## Funding

## Competing interests

The authors declare no competing interests.
