## [Peer Review File · Nature Communications]

REVIEWER COMMENTS

Reviewer #1 (Remarks to the Author):

Rereplication is often associated with cancer but the underlying mechanism is not intensively studied yet. In this manuscript, authors determined the patterns of DNA synthesis during DNA rereplication. Using genome-wide origin mapping, they demonstrated that different from firing dormant origins in cells exposed to replication stress, rereplication is initiated from the same pool of origins that are used during normal growth. This finding is interesting and important understanding the rereplication mechanism. Some points will need to be addressed to further improve the manuscript.

1. By using DNA fiber analysis, authors showed that inter-origin distance is decreased upon rereplication. In the current DNA labeling protocol (IdU 20 min/CldU 20 min), no dual-labeled signals (rereplication) were detected. It is not clear whether the origin distance measured in Fig.1G is between rereplicating origins or between normal replicating origins. As described previously (Neelsen et al., 2013), labeling cells with CldU for 2 hours, followed by IdU for 30 mins could allow monitoring fork progression and reactivation of replication origins during rereplication.
2. It is a key experiment in Fig.3 to map the origins that are used during rereplication. The results are interesting and important. However, only MLN4924 inhibitor is used, which is known to inhibit both CRL1 and CRL4. MLN4924 inhibitor may disturb pathways in addition to causing Cdt1 overexpression. Repeating origin mapping after Cdt1 overexpression would support the current findings.
3. In *Xenopus* egg extracts, Cdt1 can induce multiple rounds of DNA rereplication from the same origins (Davidson et al., 2006). It remains unknown whether rereplication in mammalian cell goes a single round or multiple rounds from the same (or adjacent) origins. With the resolution of origin mapping as shown in Fig.3B, is it possible to determine whether rereplication origin firing occurs one round or multiple rounds at certain chromosomal regions (or origins) in mammalian cells?
4. A minor point: Fig.3C demonstrated that re-replication was not accompanied by a large-scale activation of dormant origins. However, checkpoint is activated upon rereplication, which could activate dormant origins in a manner similar to that upon replication stress. Fig.3C, middle shows some difference between the samples of MLN4924 HH and HL. Could this be a result of activation of some dormant origins by checkpoint induced upon rereplication?
5. To control the specificity of Cdt1 antibody, S-phase enriched sample with low/no expression of Cdt1 should be used as a negative control for Cdt1 ChIP-seq shown in Fig.5A.
6. In the model shown in Fig.6, it is not clear whether inactive origins refer to dormant origins. If they were, they should not be activated as shown on the right side upon rereplication. Also, rereplication may also occur by refiring the same origins.

References

- Davidson, I.F., Li, A., and Blow, J.J. (2006). Deregulated replication licensing causes DNA fragmentation consistent with head-to-tail fork collision. *Mol Cell* 24, 433-443.
- Neelsen, K.J., Zanini, I.M., Mijic, S., Herrador, R., Zellweger, R., Ray Chaudhuri, A., Creavin, K.D., Blow, J.J., and Lopes, M. (2013). Deregulated origin licensing leads to chromosomal breaks by rereplication of a gapped DNA template. *Genes Dev* 27, 2537-2542.

Reviewer #2 (Remarks to the Author):

This paper examines the dynamics of DNA re-replication upon treatment with a CDT1-stabilizing drug, which is a promising treatment option for cancer. The authors find that in drug treated cancer cells, re-replication is induced, which occurs via increased initiation events at known DNA replication origins preferentially on early-replicating genomic regions. Through CDT1 Chip-seq experiments, authors suggest that re-replication at early replication domains is the result of high-density CDT1 binding to these genomic regions. While it was already published that re-replication preferentially occurs on early-replicating domains

(PMID: 32239215), many of the results of this article are novel and merit publication. The authors present well designed and well executed experiments that are analysed using correct bioinformatic methods. However, there are a few important points that need to be addressed that I believe will greatly improve the article:
(most important of which is point 4)

Major points

1. Rerep-seq (reference number 40, Menzel et al) was done in yeast and in human cells. Thus, a brief discussion of rerep-seq results in human cells and the authors experiments would be appropriate. Do authors observe signals at early replicating sub-telomeric regions? How are their regions and rerep-seq compare?
2. Re-replication "valleys" and "peaks" need to be characterized. How long are these regions? Can the authors provide Mean / median / size distribution plot. How are they distributed over the genome? How do these regions correlate with gene-rich regions?
3. It would be interesting to see how the valleys/peaks compare to TADs? Are these regions bound by TAD domains or replication domains? Is re-replication happening at the TAD borders or replication domain borders?
4. One of the most novel piece of data in this article is the CDT1-ChIP-seq. It can serve as a great resource for the field and reveals a mechanism by which re-replication takes place in response to the drug treatment. However, the methodology of CDT1 Chip-seq, its analysis and data presentation could be improved.

CDT1-ChIP-seq essentially has no controls. An IgG or preferably an "input" control is now conventional and recommended by the ENCODE (Encode data standards can be found here: https://www.encodeproject.org/chip-seq/transcription_factor/). I agree that input controls have been largely ignored by the DNA replication field, but they are essential in the correct interpretation of data in this case: "input" DNA, which is the sonicated chromatin used for immunoprecipitation, is usually enriched for open chromatin (thus early replicating DNA).

I recommend that the authors extract multiple "input" DNA controls from published ChIP-seq experiments using the same cell lines and incorporate that into Figure 5A.

Furthermore, the authors should re-call their peaks using an input DNA control to verify that their results still stand (i.e. that CDT1 peaks are enriched in early replicating domains).

In addition, the analysis represented in Figures 5B and C needs to be repeated using the peaks obtained with an input control. How many peaks per 20Kb windows do authors observe in very early, early, mid early, mid late, late and very late regions?

Similarly, the overlaps with origins (Fig 5C) need to be done directly with overlapping of origin and CDT1 binding site coordinates (I.e. using bedtools intersect). As controls, the authors need to include overlaps of origins with randomized regions of the same size and number as CDT1 peaks.

5. Phospho-MCM2 ChIP-seq also needs controls, similar to CDT1 Chip-seq. Similarly, overlaps need to be quantified: measure of direct overlaps and randomized controls.
6. A more succinct, focused discussion would benefit the reader.

Minor points

1. "The selection of active origins among the excess of prereplication complex binding sites is often tissue-specific and preferentially associates with specific chromatin modifications 13, 14, 15"

“often tissue-specific” nature of origins is probably an over-statement. In fact, reference 13 argues there are many “shared” origins. Reference 14 appears irrelevant and reference 15 is a review.

Research from Gomez, Prioleau, Aladjem and Mechali labs have addressed “constitutive/ shared / core” origins and most suggest that tissue specific origins are rare.

Further, the association with histone marks has also been addressed comprehensively by some of the aforementioned labs prior to the work cited.

Please re-phrase and please reference correctly.

2. “a methyltransferase that catalyzes the demethylation of the replication-origin-binding histone H3 on lysine 79”

Please re-phrase to be more specific; histones with this specific mark do not bind to origins (akin to transcription factor binding), rather this mark is deposited at origins or more accurately, its presence is associated with origins...

3. “These observations suggest that the mechanism underlying preferential origin activation reflects a high density of pre-replication complexes in early-replicated regions.”

Again, this concept has been proposed by Pellicci, Struhl, Prioleau and Mechali labs previously (and recently), and this needs to be correctly referenced.

Reviewer #3 (Remarks to the Author):

In the paper “Dynamics of replication origin over-activation”, the authors start from the observation that the NEDDylation inhibitor MLN4924 induces re-replication in several cell lines, through stabilisation of CDT1. Inducible over-expression of CDT1 in fact recapitulates the effects of MLN4924-treatment, comparably inducing accumulation of re-replicating cells and the activation of the DNA replication checkpoint. By devising a very elegant method, the authors show that re-replication amplifies preferentially the early replicating regions, through the activation of the same pool of origins that are normally used during unperturbed S-phase. By DNA combing, the authors find that, during re-replication, fork progression is impaired, forks are slower, increasingly asymmetric and the inter-origin distance is shorten.

The paper is very interesting and there are some very elegant experiments (such as the determination of replication distribution in the heavy-heavy fraction and identification of the origins). The key message is that MLN4924 treatment induces subsequent rounds of replication, possibly without entering G2, as the second S-phase still maintains a replication-timing program. The conclusions are intriguing, but require strengthening and a clearer model of what the authors think it is happening. The interpretation of the data could be slightly different than what I understand the authors propose. The manuscript deserves publication in Nature Communication, but first the points below need to be addressed.

1. When the second round of replication starts is a crucial question. The authors should perform a synchronised experiment, elutriating G1 cells, releasing in MLN4924 and taking time points to understand when re-replication starts. They could do a pulse of CldU and then the IdU pulse at different time points. The information from this experiment is crucial to build a model. Is replication going through the whole genome and restarting? If so, why would Cdt1 sit on the origin, inactive, until the first round of replication is finished? Or, instead, when the replication domain is completely replicated, do the origins within the domain fire again? The authors propose that re-replication does not restart immediately, as two consecutive pulses of CldU-IdU of 20 minutes each, do not show any overlap. However, replication of a domain would take about 1 hour, so, the timing of this experiment is not adequate to understand if a domain is re-replicated immediately by the end of its first round of replication.

2. The argument that the cells do not reach G2 is based solely on the fact that the replication

timing seems to be maintained. In the same experiment suggested in 1, the authors could definitely prove if the cells ever reach G2, at least in terms of cell cycle machinery. Is CycB ever stabilised? Understanding if the cell cycle machinery proceeds, although replication re-starts, is essential to build a hypothesis as why the second replication round sees such big problems with fork progression. If the authors hypothesise, for example, exhaustion of dNTPs, this would imply that the S-phase specific upregulation of dNTPs production would not be happening in the second round of S-phase. This would suggest that, at least some of the branches of the cell cycle machinery, continue towards G2 despite the stabilisation of CDT1.

3. The overlap (or lack of thereof) of the two replication rounds is important also to understand when and why the checkpoint is activated. How can the checkpoint be activated and the dormant origins not be involved? This could be explained if the checkpoint is only activated when the second round of replication starts. In this case, given the timing of the experiments, the authors would only see the beginning of the new replication round and the activation of some of the dormant origins, along with the ones that have maintained CDT1.

4. There is difference between concluding that 1. re-replication starts preferentially from the activation of early origins or 2. re-replication happens in the same order of the first round, therefore the early origins are activated, then the checkpoint is activated and the late domains never get the chance to re-replicate. I think these two possibilities should be discussed more clearly. The interesting part is the preservation of the memory of the order of replication, that either indicates, as the authors suggest, that re-replication starts before the memory is erased or that Cdt1 destruction is part of the erasure of the timing memory

5. In figure 5B, I cannot see the preferential CDT1 enrichment on the early origins during re-replication. The signal for CDT1 on the origin is blue all the way, slightly more yellow towards the late origins, because, as visible also in Fig. 5C, CDT1 is more abundant on early origins (see control cells). I think that the enrichment of CDT1 upon MLN4924 treatment is proportional to the enrichment in an unperturbed S-phase. When the second round of replication starts, stabilised CDT1 is bound proportionally to the amount of CDT1 bound during the first round of replication.

Minor points:

1. Consistency in the figure legend labelling like A and a, or (a)
2. Typo in page 9: control sample labeled with Brdu, not EdU

3. Change of font on page 25.

A detailed point-by-point response to the reviewers' comments:

Reviewer #1 (Remarks to the Author):

Rereplication is often associated with cancer but the underlying mechanism is not intensively studied yet. In this manuscript, authors determined the patterns of DNA synthesis during DNA rereplication. Using genome-wide origin mapping, they demonstrated that different from firing dormant origins in cells exposed to replication stress, rereplication is initiated from the same pool of origins that are used during normal growth. This finding is interesting and important understanding the rereplication mechanism. Some points will need to be addressed to further improve the manuscript.

Revision: We thank the reviewer for the detailed evaluation of the manuscript. The comments helped us perform a thorough revision that addresses all the points below, and we appreciate the advice.

1. By using DNA fiber analysis, authors showed that inter-origin distance is decreased upon rereplication. In the current DNA labeling protocol (IdU 20 min/CldU 20 min), no dual-labeled signals (rereplication) were detected. It is not clear whether the origin distance measured in Fig.1G is between rereplicating origins or between normal replicating origins. As described previously (Neelsen et al., 2013), labeling cells with CldU for 2 hours, followed by IdU for 30 mins could allow monitoring fork progression and reactivation of replication origins during rereplication.

Revision: We agree with the reviewer that the temporal separation we have observed is non-obvious and merits further investigation. As reported in the revised manuscript, we have measured the rates of re-replication by labeling cells for a longer time. We have not observed dual labels when we used the exact labeling protocol (2 hours followed by 30 minutes) reported by Neelsen et al. (ref. 37 in the revised paper) in MLN4924 treated, but dual labels were detected after longer labeling periods (Supplementary Figure 2, discussed on page 8, first paragraph). These observations support the conclusion that re-replication did not initiate immediately after the first round of DNA synthesis; rather, the second round of replication begins after a substantial portion of early-replicating chromatin had already been duplicated. These observations are now discussed in the revised manuscript (for example page 19, last paragraph).

2. It is a key experiment in Fig.3 to map the origins that are used during rereplication. The results are interesting and important. However, only MLN4924 inhibitor is used, which is known to inhibit both CRL1 and CRL4. MLN4924 inhibitor may disturb pathways in addition to causing Cdt1 overexpression. Repeating origin mapping after Cdt1 overexpression would support the current findings.

Revision: We thank the reviewer for this important suggestion. As suggested, in the revision we include data mapping replication origins in Cdt1 overexpressing cells. As shown in Supplementary Figure 6, d and e, re-replication in those cells also utilized the same set of origins as in normally replicating cells, as shown for cells treated with MLN4924. All the data from origin mapping in CDT1 overexpressing cells will also be made public in GEO upon publication of the paper.

3. In *Xenopus* egg extracts, Cdt1 can induce multiple rounds of DNA rereplication from the same origins (Davidson et al., 2006). It remains unknown whether rereplication in mammalian cell goes a single round or multiple rounds from the same (or adjacent) origins. With the resolution of origin mapping as shown in Fig.3B, is it possible to determine whether rereplication origin firing occurs one round or multiple rounds at certain chromosomal regions (or origins) in mammalian cells?

Revision: Thank you, we agree, this is a very interesting point. Our high-resolution copy number based mapping of replication origins in re-replicating cells suggest that cells are capable of initiating multiple rounds of replication as shown by Davidson et al. (ref. 38 in the revised submission) for *Xenopus* egg extract, and further demonstrate that these multiple rounds of initiation preferentially occur at a subset of early origins. Based on the reviewer's comment, we have mentioned and discussed this point in the revised version (Results page 14, first paragraph, Discussion, paragraph spanning pages 17-18).

4. A minor point: Fig.3C demonstrated that re-replication was not accompanied by a large-scale activation of dormant origins. However, checkpoint is activated upon rereplication, which could activate dormant origins in a manner similar to that upon replication stress. Fig.3C, middle shows some difference between the samples of MLN4924 HH and HL. Could this be a result of activation of some dormant origins by checkpoint induced upon rereplication?

Revision: Yes, we agree and we thank the reviewer for mentioning this important consideration. Our observations suggest that re-replication occurs primarily at the origins that are used during routine replication, albeit at a higher frequency associated with the slowing of replication fork movement. We expect that such origins will be at early replicating regions and indeed we observed this in Figure 4d. However, we did not detect frequent dormant origins although some activation of dormant origins is expected to accompany replication perturbations and DNA damage, especially as cells experience prolonged rereplication. This point is discussed in the submitted revised manuscript (page 19 second paragraph).

5. To control the specificity of Cdt1 antibody, S-phase enriched sample with low/no expression of Cdt1 should be used as a negative control for Cdt1 ChIP-seq shown in Fig.5A.

Revision: We thank the reviewer for bringing up this important control. In the revision, we have included a ChIP-Seq control from cells in mid-late S-phase, which do not express CDT1, indeed

demonstrating the specificity of the immunoprecipitation. These data (track labeled “MS input” are shown in Supplementary Figure 9a.

6. In the model shown in Fig.6, it is not clear whether inactive origins refer to dormant origins. If they were, they should not be activated as shown on the right side upon rereplication. Also, rereplication may also occur by refiring the same origins.

Revision: We thank the reviewer and we agree. We have redrawn the model to clarify the designation of inactive origins and active origins including dormant origins, frequent origins and intermittent origins.

Reviewer #2 (Remarks to the Author):

This paper examines the dynamics of DNA re-replication upon treatment with a CDT1-stabilizing drug, which is a promising treatment option for cancer. The authors find that in drug treated cancer cells, re-replication is induced, which occurs via increased initiation events at known DNA replication origins preferentially on early-replicating genomic regions.

Through CDT1 Chip-seq experiments, authors suggest that re-replication at early replication domains is the result of high-density CDT1 binding to these genomic regions.

While it was already published that re-replication preferentially occurs on early-replicating domains (PMID: 32239215), many of the results of this article are novel and merit publication.

The authors present well designed and well executed experiments that are analyzed using correct bioinformatic methods. However, there are a few important points that need to be addressed that I believe will greatly improve the article:

(most important of which is point 4)

Revision: We thank the reviewer for the thorough evaluation of the manuscript. We were happy to learn that the reviewer considers our observations novel and utilize correct methodologies. We agree that the results are in line with the preference for early-replicating regions shown in the simulation of re-replication performed by Menzel et al, 2020 (as cited in our manuscript – original reference #40, reference # 42 in the revision). Our analyses of the mapped origin utilization in re-replicating cells (not performed by Menzel et al.) are also in line with the preference for early replicating regions. Below we provide a detailed description of how we addressed all the important points raised by the reviewer:

Major points

1. Rerep-seq (reference number 40, Menzel et al) was done in yeast and in human cells. Thus, a brief discussion of rerep-seq results in human cells and the authors experiments would be appropriate. Do authors observe signals at early replicating sub-telomeric regions? How are their regions and rerep-seq compare?

Revision: We thank the reviewer for this excellent suggestion, a discussion of this point in the context of the Menzel et al. report (ref. 42 in the revised submission) is appropriate and is included in the revision (page 9, last sentence; page 17, second paragraph). It should be noted that direct comparisons with the Menzel et al., data are limited by two caveats: First, because re-rep-seq delineated large replication domains but did not directly map replication origins, therefore direct evaluations of co-localizations were not possible. Second, mapping replication origins in sub-telomeric regions is beyond the capabilities of our method, which cannot map repetitive regions. Such mapping will probably be possible with more advanced, longer-read sequencing in the future.

2. Re-replication “valleys” and “peaks” need to be characterized. How long are these regions? Can the authors provide Mean / median / size distribution plot. How are they distributed over the genome? How do these regions correlate with gene-rich regions?

Revision: We thank the reviewer for drawing our attention to this issue. We replaced “valleys” and “peaks” with “Low” and “High” to indicate peak height to simplify the texts. In the revised version we provide the size distribution of these regions (Supplementary Figure 5A). Our data demonstrate that indeed, as the reviewer had anticipated, the peak regions correlate with gene-rich regions (Supplementary Figure 5b). We also expand the analyses to include correlations with chromatin compartments (correlations with Hi-C Eigen scores, Figure 2,c and e, Supplementary Figure 5c) suggesting a strong association with euchromatin, compartment A.

3. It would be interesting to see how the valleys/peaks compare to TADs? Are these regions bound by TAD domains or replication domains? Is re-replication happening at the TAD borders or replication domain borders?

Revision: We thank the reviewer for this interesting suggestion. Our analyses suggest that the broad peaks we have observed were highly correlated ($R=0.73$) with euchromatic regions and associated with early replication timing domains, but not with TADs (for example, see the screenshot in Figure 2c). Correlations with chromatin modifications associated with euchromatin and heterochromatin corroborate these conclusions. These analyses are shown in Figure 2, c-e and Supplementary Table 2 (discussion: page 10, second paragraph) .

4. One of the most novel piece of data in this article is the CDT1-ChIP-seq. It can serve as a great resource for the field and reveals a mechanism by which re-replication takes place in response to the drug treatment. However, the methodology of CDT1 Chip-seq, its analysis and data presentation could be improved.

CDT1-ChIP-seq essentially has no controls. An IgG or preferably an “input” control is now conventional and recommended by the ENCODE (Encode data standards can be found here: https://www.encodeproject.org/chip-seq/transcription_factor/). I agree that input controls have been largely ignored by the DNA replication field, but they are essential in the correct interpretation of data in this case: “input” DNA, which is the sonicated chromatin used for immunoprecipitation, is usually enriched for open chromatin (thus early replicating DNA).

I recommend that the authors extract multiple “input” DNA controls from published CHIP-seq experiments using the same cell lines and incorporate that into Figure 5A.

Revision: We thank the reviewer for pointing out this omission, we apologize for not showing the standard controls we have used in the previous version of the manuscript. In the revised submission, we have included the control tracks according to the ENCODE guidelines suggested by the reviewer. Each experiment included a corresponding input control with a matching run type, read length and replicate structure that was used in peak calling and the analyses shown in Figure 5 and Supplementary Figure 9. The entire dataset, including inputs, will be publicly accessible in GEO upon publication.

In addition, the analysis represented in Figures 5B and C needs to be repeated using the peaks obtained with an input control .

Revision: We thank the reviewer for the suggestion, peaks were called against an input control as suggested by the reviewer. In the revised manuscript the input controls are included and the analyses are represented as the reviewer suggested (Figure 5, Supplementary Figure 9).

How many peaks per 20Kb windows do authors observe in very early, early, mid early, mid late, late and very late regions?

Revision: The distribution of peaks in the replication timing domains listed above is shown in Figure 5c. We have chosen to use FPKM instead of peaks per 20 kb for ease of presentation.

Similarly, the overlaps with origins (Fig 5C) need to be done directly with overlapping of origin and CDT1 binding site coordinates (I.e. using bedtools intersect). As controls, the authors need to include overlaps of origins with randomized regions of the same size and number as CDT1 peaks.

Revision: Thank you for this excellent suggestion. We have used bedtools intersect to compare origin and CDT1 binding sites directly using randomized regions as controls. The results are shown in Supplementary Figure 9c and a similar analysis for phosphorylated MCM2 in Supplementary Figure 10d.

5. Phospho-MCM2 CHIP-seq also needs controls, similar to CDT1 Chip-seq. Similarly, overlaps need to be quantified: measure of direct overlaps and randomized controls.

Revision: We thank the reviewer again; the controls were included similar to the above and overlaps were calculated. The results are shown in Supplementary Figure 10.

6. A more succinct, focused discussion would benefit the reader.

Revision: We have shortened the discussion by 20% by eliminating redundancies, while addressing some additional issues as pointed out by the reviewers.

Minor points:

1. “The selection of active origins among the excess of prereplication complex binding sites is often tissue-specific and preferentially associates with specific chromatin modifications 13, 14, 15, “often tissue-specific” nature of origins is probably an over-statement. In fact, reference 13 argues there are many “shared” origins. Reference 14 appears irrelevant and reference 15 is a review.

Research from Gomez, Prioleau, Aladjem and Mechali labs have addressed “constitutive/ shared / core” origins and most suggest that tissue specific origins are rare. Further, the association with histone marks has also been addressed comprehensively by some of the aforementioned labs prior to the work cited. Please re-phrase and please reference correctly.

Revision: We thank the reviewer for noticing the issue. The sentence was rephrased, the terms were clarified and references were updated.

2. “a methyltransferase that catalyzes the demethylation of the replication-origin-binding histone H3 on lysine 79”

Please re-phrase to be more specific; histones with this specific mark do not bind to origins (akin to transcription factor binding), rather this mark is deposited at origins or more accurately, its presence is associated with origins...

Revision: thank you, we agree. The sentence was revised to state that H3K79Me2 was associated with, not binding, replication origins.

3. “These observations suggest that the mechanism underlying preferential origin activation reflects a high density of pre-replication complexes in early-replicated regions.”

Again, this concept has been proposed by Pellicci, Struhl, Prioleau and Mechali labs previously (and recently), and this needs to be correctly referenced.

Revision: Thank you, relevant studies/reviews from the above laboratories were cited.

Reviewer #3 (Remarks to the Author):

In the paper “Dynamics of replication origin over-activation”, the authors start from the observation that the NEDDylation inhibitor MLN4924 induces re-replication in several cell lines, through stabilisation of CDT1. Inducible over-expression of CDT1 in fact recapitulates the effects of MLN4924-treatment, comparably inducing accumulation of re-replicating cells and

the activation of the DNA replication checkpoint. By devising a very elegant method, the authors show that re-replication amplifies preferentially the early replicating regions, through the activation of the same pool of origins that are normally used during unperturbed S-phase. By DNA combing, the authors find that, during re-replication, fork progression is impaired, forks are slower, increasingly asymmetric and the inter-origin distance is shortened.

The paper is very interesting and there are some very elegant experiments (such as the determination of replication distribution in the heavy-heavy fraction and identification of the origins). The key message is that MLN4924 treatment induces subsequent rounds of replication, possibly without entering G2, as the second S-phase still maintains a replication-timing program. The conclusions are intriguing, but require strengthening and a clearer model of what the authors think it is happening. The interpretation of the data could be slightly different than what I understand the authors propose. The manuscript deserves publication in Nature Communication, but first the points below need to be addressed.

Revision: We thank the reviewer for the thorough evaluation and the helpful suggestions. We are happy to learn that the reviewer thinks that the experiments we present are elegant and that our results are intriguing. Below we detail how we have addressed all the points raised by the reviewer.

1. When the second round of replication starts is a crucial question. The authors should perform a synchronised experiment, elutriating G1 cells, releasing in MLN4924 and taking time points to understand when re-replication starts. They could do a pulse of CldU and then the IdU pulse at different time points. The information from this experiment is crucial to build a model.

Is replication going through the whole genome and restarting? If so, why would Cdt1 sit on the origin, inactive, until the first round of replication is finished? Or, instead, when the replication domain is completely replicated, do the origins within the domain fire again?

The authors propose that re-replication does not restart immediately, as two consecutive pulses of CldU-IdU of 20 minutes each, do not show any overlap. However, replication of a domain would take about 1 hour, so, the timing of this experiment is not adequate to understand if a domain is re-replicated immediately by the end of its first round of replication.

Revision: We thank the reviewer for the suggestion, this is indeed an interesting question. As suggested, we have performed a time course of re-replication with synchronized MLN4924 treated cells and the results (Supplementary Figure 1g) suggest that cells begin re-replication towards the end of S-phase. We have also determined the abundance of S-phase markers (cyclin B and phospho-histone-H3 ser10) to determine more accurately the cell cycle status at the onset of re-replication. These data are shown in Supplementary Figures 1, e and f.

Additional information about the onset of re-replication, also suggesting a lag between the first round of replication and the initiation of re-replication, can be deduced from the experiments combining experiments described in the response to reviewer #1, point 1 (supplementary Figure 2), and from the replication timing data demonstrating a strong preference for early origins (Figure 4).

2. The argument that the cells do not reach G2 is based solely on the fact that the replication timing seems to be maintained. In the same experiment suggested in 1, the authors could definitely prove if the cells ever reach G2, at least in terms of cell cycle machinery. Is CycB ever stabilised?

Understanding if the cell cycle machinery proceeds, although replication re-starts, is essential to build a hypothesis as why the second replication round sees such big problems with fork progression. If the authors hypothesise, for example, exhaustion of dNTPs, this would imply that the S-phase specific upregulation of dNTPs production would not be happening in the second round of S-phase. This would suggest that, at least some of the branches of the cell cycle machinery, continue towards G2 despite the stabilisation of CDT1.

Revision: We thank the reviewer for this suggestion, which prompted us to analyze cyclin B in the re-replicating cells. Cyclin B levels are high in re-replicating cells, as they increase at the end of S-phase during routine cell cycle progression, and we do not observe cyclin B1 degradation. These analyses indeed demonstrated that cells initiate the second round of replication while still in S-phase - the later-replicating portion of the genome is still undergoing the first round of duplication. In the revised submission, we include cyclin B data as well as measurements of phosphorylated histone H3 Ser10, substantiating this point (supplementary Figure 1, e and f).

3. The overlap (or lack of thereof) of the two replication rounds is important also to understand when and why the checkpoint is activated. How can the checkpoint be activated and the dormant origins not be involved? This could be explained if the checkpoint is only activated when the second round of replication starts. In this case, given the timing of the experiments, the authors would only see the beginning of the new replication round and the activation of some of the dormant origins, along with the ones that have maintained CDT1.

Revision: We thank the reviewer for raising this point. Our single fiber analyses (Supplementary Figure 2) support a model whereby the replicon located within early-replicating chromatin domains complete their duplication before the second round of replication begins. In re-replicating cells, recruitment of additional pre-replication complexes (that normally is prohibited by CDT1 degradation) initiates a second round of DNA synthesis on the early replicons without completing DNA synthesis in the late-replicating portion of the genome. As suggested by the reviewer, it is possible that replication perturbations accompanying re-replication can lead to checkpoint activation during, and not before, the second round of replication. As also pointed out by reviewer #1, although our results show that most re-replication events initiate from origins that are also utilized during routine genome duplication,

we cannot rule out some activation of dormant origins. These issues are discussed in the revised submission (page 19, second paragraph).

Minor points:

1. Consistency in the figure legend labelling like A and a, or (a)
2. Typo in page 9: control sample labeled with Brdu, not EdU
3. Change of font on page 25.

Revision: corrected.

In sum, we thank all the reviewers for the thorough reading and the thoughtful comments. The reviewers' suggestions had significantly improved the manuscript and we appreciate their time and help.

REVIEWERS' COMMENTS

Reviewer #1 (Remarks to the Author):

The authors have nicely addressed the reviewers' comments. This revised version will be an important contribution to the understanding of DNA rereplication in mammalian cells. It will be of great interest to the broad readership of Nature Communications.

Reviewer #2 (Remarks to the Author):

The authors have satisfactorily responded to my queries. I would be looking forward to reading it at Nature Comm.

Of note, on point 2: I did not mean to ask the authors to rename peaks/valleys, just to characterise it further. I was happy with their terminology, which was in line with the literature.

Reviewer #3 (Remarks to the Author):

I am happy with the revision and I find the data about cyclin B1 accumulation in absence of phosphorylated Ser 10 histone H3 very intriguing. I am looking forward to see the manuscript published.

A detailed point-by-point response to the reviewers' comments:

Reviewer #1 (Remarks to the Author):

The authors have nicely addressed the reviewers' comments. This revised version will be an important contribution to the understanding of DNA rereplication in mammalian cells. It will be of great interest to the broad readership of Nature Communications.

Response: We thank the reviewer for the comment and for the evaluation of the revision.

Reviewer #2 (Remarks to the Author):

The authors have satisfactorily responded to my queries. I would be looking forward to reading it at Nature Comm.

Of note, on point 2: I did not mean to ask the authors to rename peaks/valleys, just to characterise it further. I was happy with their terminology, which was in line with the literature.

Response: We thank the reviewer for the comment and for the advice during the review process. We are happy with the revised terminology and are glad that the reviewer was satisfied with the additional characterization.

Reviewer #3 (Remarks to the Author):

I am happy with the revision and I find the data about cyclin B1 accumulation in absence of phosphorylated Ser 10 histone H3 very intriguing. I am looking forward to see the manuscript published.

Response: We thank the reviewer for the comment and for the evaluation of the new data.

In sum, we thank all the reviewers for the thoughtful comments during the initial review and the reevision. The reviewers' suggestions have significantly improved the manuscript and we appreciate their time and help.